# Molecular characterization of gustatory second-order neurons reveals integrative mechanisms of gustatory and metabolic information

**Rubén Mollá-Albaladejo, Manuel Jiménez-Caballero, Juan Antonio Sanchez-Alcaniz***

Instituto de Neurociencias, CSIC-UMH, San Juan de Alicante, Spain

## eLife Assessment

This study provides **valuable** insights into the organization of second-order circuits of gustatory neurons, particularly in how these circuits integrate opposing taste inputs and are modulated by metabolic state to regulate feeding behavior. Through an elegant combination of complementary techniques, the authors identify the target neurons involved in gustatory integration. The evidence supporting their conclusions is **convincing**.

***For correspondence:**
juan.sanchez@umh.es

**Competing interest:** The authors declare that no competing interests exist.

**Abstract** Animals must balance the urgent need to find food during starvation with the critical necessity to avoid toxic substances to ensure their survival. In *Drosophila*, specialized Gustatory Receptors (GRs) expressed in Gustatory Receptor Neurons (GRNs) are critical for distinguishing between nutritious and potentially toxic food. GRNs project their axons from taste organs to the Subesophageal Zone (SEZ) in the Central Brain (CB) of *Drosophila*, where gustatory information is processed. Although the roles of GRs and GRNs are well-documented, the processing of gustatory information in the SEZ remains unclear. To better understand gustatory sensory processing and feeding decision-making, we molecularly characterized the first layer of gustatory interneurons, referred to as Gustatory Second-Order Neurons (G2Ns), which receive direct input from GRNs. Using trans-synaptic tracing with *trans*-Tango, cell sorting, and bulk RNAseq under fed and starved conditions, we discovered that G2Ns vary based on gustatory input and that their molecular profile changes with the fly's metabolic state. Further data analysis has revealed that a pair of neurons in the SEZ, expressing the neuropeptide Leucokinin (SELK neurons), receive simultaneous input from GRNs sensing bitter (potentially toxic) and sweet (nutritious) information. Additionally, these neurons also receive inputs regarding the starvation levels of the fly. These results highlight a novel mechanism of feeding regulation and metabolic integration.

## Introduction

Animals face constant dangers in their environment that they must avoid to survive, with starvation and poisoning two critical threats. To prevent death by starvation, animals must continually seek food, even if it means risking consumption of potentially harmful substances (*Chu et al., 2014*; *Inagaki et al., 2014*; *LeDue et al., 2016*). Therefore, accepting or rejecting potentially poisonous food will depend on the general nutritious content and toxicity of the meal, together with the starvation level of the animal. Similar to other animals, *Drosophila melanogaster* can discern different taste qualities (sweet, bitter, salty, sour, umami, or carbonation) through various populations of GRNs tuned

to each of these qualities (*Yarmolinsky et al., 2016*). GRNs are distributed across the *Drosophila*'s body, including the proboscis, legs, wing margins, and the ovipositor organ, expressing a combination of receptors for detecting various food compounds (*Vosshall and Stocker, 2007*). There are four major families of receptors involved in gustatory perception: GRs, Ionotropic Receptors (IRs), Transient Receptor Proteins (TRPs), and pickpocket (ppk) channels (*Montell, 2021*). Specific GRNs are associated with particular valences, such as sweet or bitter tastes. For instance, Gr64f is a receptor for sugar detection promoting feeding behavior (*Dahanukar et al., 2007*; *Fujii et al., 2015*), while Gr66a is present in GRNs that detect bitter compounds, leading to feeding rejection (*Dunipace et al., 2001*; *Wang et al., 2004*).

GRNs project their axons to the SEZ, the main brain region in flies for integrating and processing gustatory information. Unlike the olfactory system, where olfactory sensory neurons expressing the same receptors converge in the same glomeruli to synapse with projection neurons, the gustatory system lacks such anatomical subdivisions (*Vosshall and Stocker, 2007*; *Scott, 2018*). However, neurons expressing receptors that respond to sweet compounds and elicit attractive behaviors converge in specific SEZ regions without overlapping neurons detecting bitter compounds and inducing feeding rejection (*Wang et al., 2004*; *Thorne et al., 2004*).

The development of extensive collections of Gal4 lines (*Jenett et al., 2012*), split-Gal4 collections (*Dionne et al., 2018*), and databases for searchable neurons (*Meissner et al., 2023*) has facilitated the identification of gustatory interneurons in the SEZ involved in feeding behavior. Those databases have been used to design experimental procedures that have helped in the identification of some gustatory interneurons involved in integrating gustatory information and inducing feeding (*Chu et al., 2014*; *Flood et al., 2013*; *Kain and Dahanukar, 2015*; *Miyazaki et al., 2015*; *Yapici et al., 2016*; *Kim et al., 2017*; *Schwarz, 2017*; *Bohra et al., 2018*). These studies have identified command neurons critical for feeding behavior, some directly receiving input from GRNs, such as certain GABAergic neurons (*Chu et al., 2014*), while others, like the Feeding neuron (Fdg), act as a hub to control food acceptance without being connected directly to any GRN (*Flood et al., 2013*). Despite these advances, the role of gustatory interneurons in integrating gustatory information and the metabolic state of the fly remains largely unexplored. The recent development of the *Drosophila* brain connectome through electron microscopy is advancing our understanding of SEZ connectivity and the complex interactions among GRNs (*Sterne et al., 2021*; *Engert et al., 2022*).

We have aimed to elucidate the integration of gustatory information and the metabolic state, by analyzing the first layer of interneurons that collect gustatory information, G2Ns (*Sterne et al., 2021*). We have decided to focus on understanding how flies integrate gustatory information that codes for opposing behaviors: sweet tastants elicit feeding, whereas bitter compounds induce food rejection (*Scott, 2018*). We have followed a molecular approach to identify G2Ns. First, we employed the *trans*-Tango genetic tool (*Talay et al., 2017*), which labels postsynaptic neurons connected to specific presynaptic neurons. Later, using Fluorescent Activated Cell Sorting (FACS) of labeled G2Ns and bulk RNA sequencing (RNAseq), we characterized the molecular profiles of postsynaptic neurons connected to sweet and bitter GRNs in fed and starved conditions for the first time. These analyses show that G2Ns receive input from molecularly distinct GRNs, and their transcriptional profiles change upon starvation, presumably to adapt to the new metabolic state. Using molecular and connectomic techniques, we further demonstrate that the two *Leucokinin*-expressing neurons in the SEZ (SELK neurons) are G2Ns that simultaneously receive sweet and bitter GRN input. Behavioral experiments revealed that Leucokinin neurons integrate information regarding the metabolic state of the fly and are involved in the processing of bitter and sweet information that impacts the initiation of feeding behavior.

## Results

### Molecular characterization of gustatory second-order neurons

We employed the trans-synaptic tracing method *trans*-Tango (*Talay et al., 2017*), to label G2Ns synaptically connected to sweet and bitter GRNs. To label G2Ns receiving sweet information, we used the *Gr64f-Gal4* transgene (*Dahanukar et al., 2007*; *Wang et al., 2004*), and for G2Ns receiving bitter information, we used the *Gr66a-Gal4* transgene (*Wang et al., 2004*; *Weiss et al., 2011*). These Gal4 transgenes enabled us to label multiple G2Ns of interest (*Figure 1—figure supplement 1A and B*).

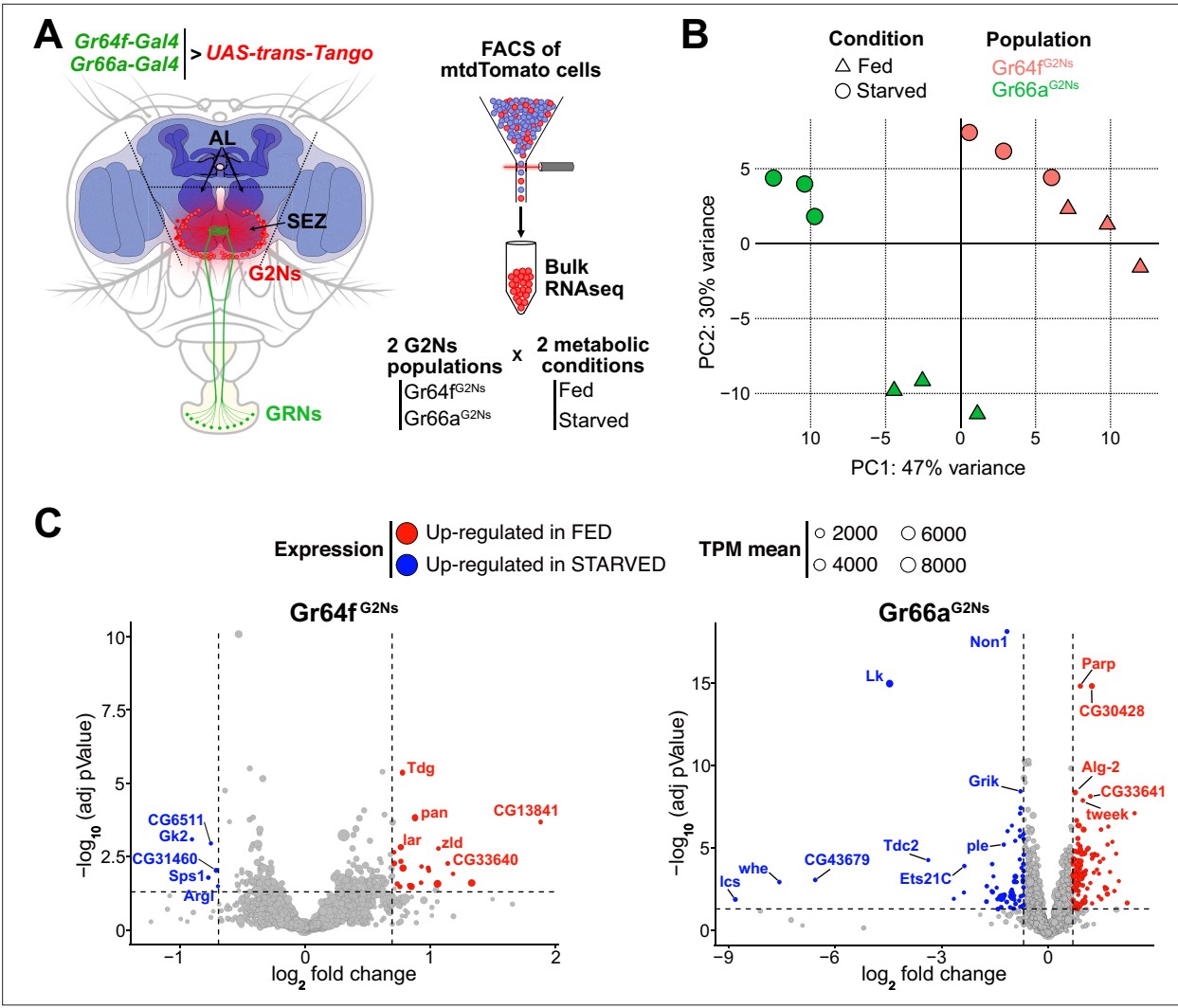

**Figure 1.** The metabolic state affects the gene profiles of the Gustatory Second-Order Neurons (G2Ns) populations analyzed. (**A**) The scheme depicts the region of the brain dissected and the sorting of the G2Ns according to the input. Two populations of *trans*-Tango neurons were labeled using *Gr64f-Gal4* and *Gr66a-Gal4* transgenic lines. Flies were either fed or food-starved for 24 hr. (**B**) Principal Component Analysis (PCA) of the genes. (**C**) Volcano plots show the genes up- and down-regulated in fed and starved conditions in the two populations of G2Ns.

The online version of this article includes the following figure supplement(s) for figure 1:

**Figure supplement 1.** Variation in Gustatory Second-Order Neuron (G2N) numbers for the different populations under study.

**Figure supplement 2.** Gene Ontology (GO) analysis of the genes differentially expressed in fed and starved conditions for the two Gustatory Second-Order Neurons (G2Ns) populations analyzed.

The number of postsynaptic G2Ns differed between the GRNs, indicating differences in information processing for different tastants (*Figure 1—figure supplement 1C and D*). Since satiety and hunger significantly influence behavior, modifying olfactory behavior (*Root et al., 2011*), memory formation (*Krashes et al., 2009*) and food consumption (*Inagaki et al., 2014*; *Riemensperger et al., 2011*), we extended our analysis to characterize molecularly G2Ns receiving different gustatory inputs in two metabolic states: fed and 24 hr starved flies. Given that all G2Ns are fluorescently labeled with mtdTomato, we used FACS to separate and collect the cells for bitter and sweet G2N populations (Gr66a^G2Ns and Gr64f^G2Ns, respectively) in the two metabolic conditions (see Materials and methods for a detailed description of the sorting procedure and sample number in *Supplementary file 1*). We then performed bulk RNAseq for each condition (*Figure 1A*).

The data's Principal Component Analysis (PCA) revealed that the transcriptomic profiles of the two G2N populations are different and that the metabolic state (fed vs. starved) significantly influences

gene expression without affecting the separation among the populations in the PC space (*Figure 1B*). These findings suggest that the first layer of neurons collecting sensory information is already modulated by the fly's metabolic state, potentially impacting subsequent information processing.

We further analyzed the changes in gene expression between fed and starved conditions across the two G2N populations. Many genes exhibited differential expression in starved flies compared to fed flies (*Figure 1C*). Gene Ontology (GO) analysis indicated that several genes were involved in nervous system development and synapse organization, suggesting the importance of synapse remodeling during starvation adaptation. Additionally, GO terms related to the response to sensory stimuli and stress response were identified, implying that starvation induces molecular signatures of stress in flies. Finally, some genes associated with signaling pathways indicated that neurotransmitter and neuropeptide signaling processes might be altered during starvation, leading to specific adaptations in the neuronal circuits involving G2Ns (*Figure 1—figure supplement 2*).

## Characterization of neurotransmitter and neuropeptide expression

Next, we focused on the expression of neurotransmitters, neuropeptides, and their respective receptors as they are known to modulate feeding among many other behaviors (*Nässel and Zandawala, 2019*). For example, the CCHamide 1 receptor regulates starvation-induced olfactory modulation *Farhan et al., 2013*; NPF (Neuropeptide F) modulates taste perception in starved flies by increasing sensitivity to sugar and decreasing sensitivity to bitter by promoting GABAergic inhibition onto bitter GRNs *Inagaki et al., 2014*; starvation enhances behavioral sugar-sensitivity via increased dopamine release onto sweet GRNs, indirectly decreasing sensitivity to bitter tastants *Inagaki et al., 2012*; and octopamine signaling affects bitter signaling in starved flies (*LeDue et al., 2016*).

Our RNAseq data revealed that the most prominent neurotransmitter receptor expressed in all G2Ns was *nAChRbeta1* (nicotinic Acetylcholine Receptor Beta 1) (*Figure 2A*). This is consistent with previous findings showing that GRNs are mostly cholinergic (*Jaeger et al., 2018*). Other acetylcholine and neurotransmitter receptors were also expressed in G2Ns. This indicates that while G2Ns receive significant input from GRNs, they also integrate information from a complex network of other neurons using different neurotransmitters. We found no variation in expression levels for any neurotransmitter receptors when comparing fed and starved conditions. Regarding neurotransmitter expression, *Gad1* (Glutamic acid decarboxylase 1) and *Ddc* (Dopa decarboxylase), two essential enzymes involved in the synthesis of GABA and dopamine, respectively, were highly expressed in the two G2N populations for both metabolic states (*Figure 2B*). Other enzymes involved in the GABA pathway, like *VGAT* (vesicular GABA transporter) or *GABAT* (γ-aminobutyric acid transaminase), were also expressed but at lower levels than *Gad1*. Similarly, other enzymes or transporters related to other neurotransmitters, like glutamate (Glu) or serotonin (5-HT), were present but at lower levels.

Regarding neuropeptide receptors, we found many receptors expressed in all G2Ns populations with varying expression levels (*Figure 2C*). For instance, the *EcR* (Ecdyson Receptor) showed expression in all three populations with a slight decrease under starvation, indicating a possible role in adapting to food deprivation. Additionally, *sNPF-R* (short Neuropeptide F Receptor) was highly expressed with similar expression patterns as its ligand sNPF (short Neuropeptide F), highlighting the importance of sNPF and its signaling pathway in regulating starvation and feeding (*Lee et al., 2004*; *Wu et al., 2003*; *Yoshinari et al., 2021*).

Neuropeptides showed variable expression patterns between populations and conditions (*Figure 2D*). For example, *sNPF* was highly expressed in all G2Ns, particularly in the Gr66a[G2N] population, without changes associated with the metabolic state. sNPF production, which is related to circulating sucrose levels, impacts food intake by integrating sweet and bitter information in food-deprived flies (*Inagaki et al., 2014*). This highlights the role of sNPF and its receptor (sNPF-R) in regulating the feeding behavior according to the metabolic state, as it is secreted in the midgut and by neurosecretory cells in the brain. Dh31 (Diuretic hormone 31) and Dh44 (Diuretic hormone 44), which are involved in desiccation tolerance and starvation adaptation (*Cannell et al., 2016*), also showed notable expression in both G2N populations under starved and fed conditions. Nplp1 (Neuropeptide-like precursor 1), which is associated with synaptic organization in the nervous system and water balance (*Nässel and Zandawala, 2019*), was also expressed in both G2Ns. Among all neuropeptides, however, Leucokinin (Lk) displayed the most striking expression change. Though *Lk* has no (or very low) expression in the Gr64f[G2N] population under fed and starved conditions, and no (or very low) expression in the

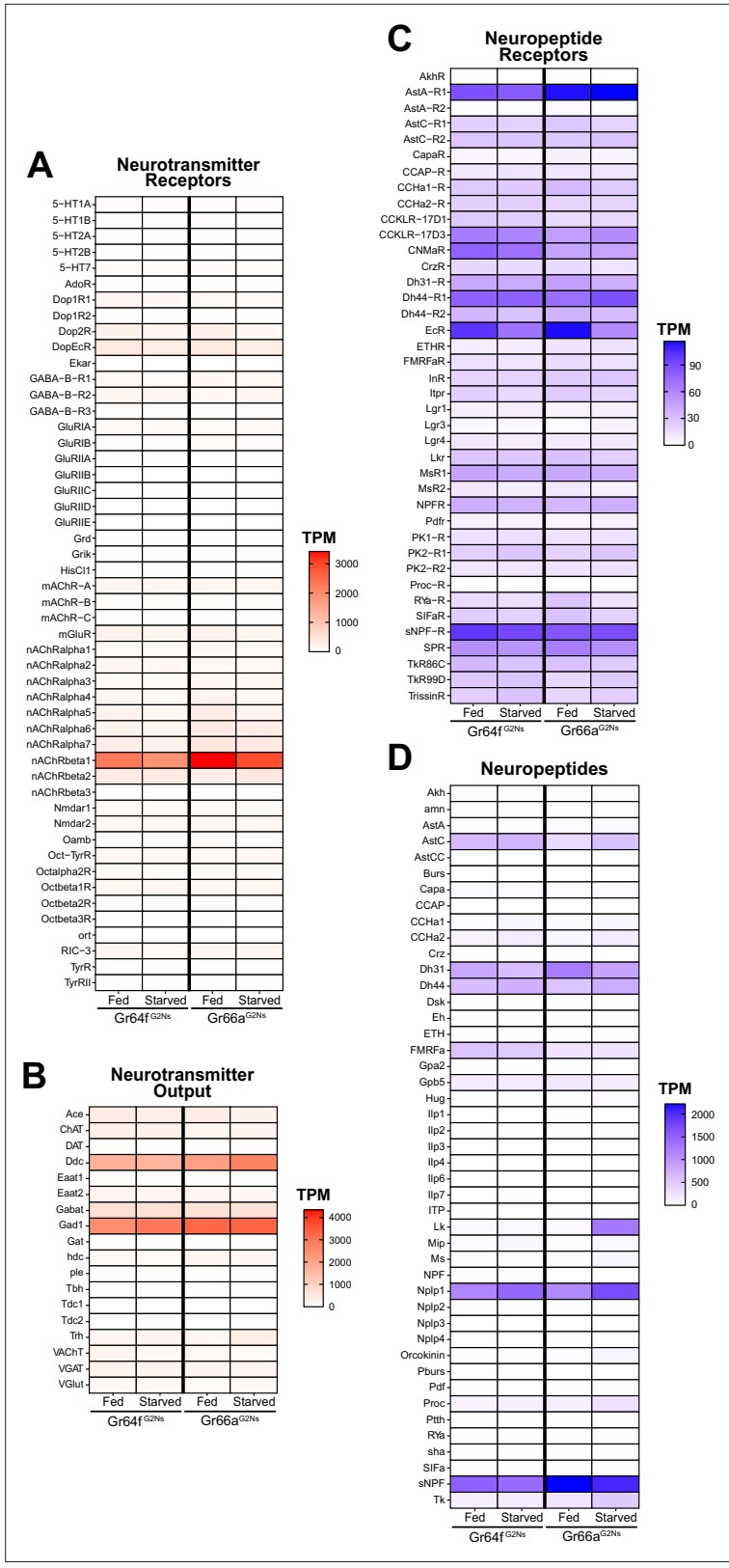

**Figure 2.** Transcripts per Million (TPM) for Neuropeptides and Neurotransmitters. TPMs for Neurotransmitters (**A**), Neurotransmitter Receptors (**B**), Neuropeptides (**C**), and Neuropeptide Receptors (**D**) for the two Gustatory Second-Order Neuron (G2N) populations in the two metabolic conditions studied (fed and 24 hr starved).

Gr66a[G2N] under fed conditions, *Lk* showed significantly elevated expression in the Gr66a[G2N] population under starved conditions. This expression pattern specific to Gr66a[G2Ns] suggests that Lk may play a role in processing bitter information under starvation. We decided to explore this neuropeptide further to understand its role in integrating feeding behavior and the metabolic state of the flies.

## Leucokinin expression is upregulated SELK neurons in the starved condition

Leucokinin is a neuropeptide with multiple roles in various physiological and behavioral processes (*Nässel, 2021*). Three different Lk neuron groups exist in the adult *Drosophila* Central Nervous System (CNS). There are approximately 20 Lk neurons in the Ventral Nerve Cord (ABLKs), while in the CB, there are four Lk neurons, two located in the Lateral Horn (LHLK) and another two located in the SEZ (SELK) (*de Haro et al., 2010*; *Zandawala et al., 2018*). ABLKs use Lk as a hormonal signal targeting peripheral tissues, including renal tubules, to regulate water and ion homeostasis in response to desiccation, starvation, and ion stress, as Lk regulates fluid secretion in the Malpighian (renal) tubules (*Zandawala et al., 2018*). LHLK neurons are part of the output circuitry of the circadian clock, regulating locomotor activity and sleep suppression induced by starvation (*Al-Anzi et al., 2010*; *Murakami et al.,*

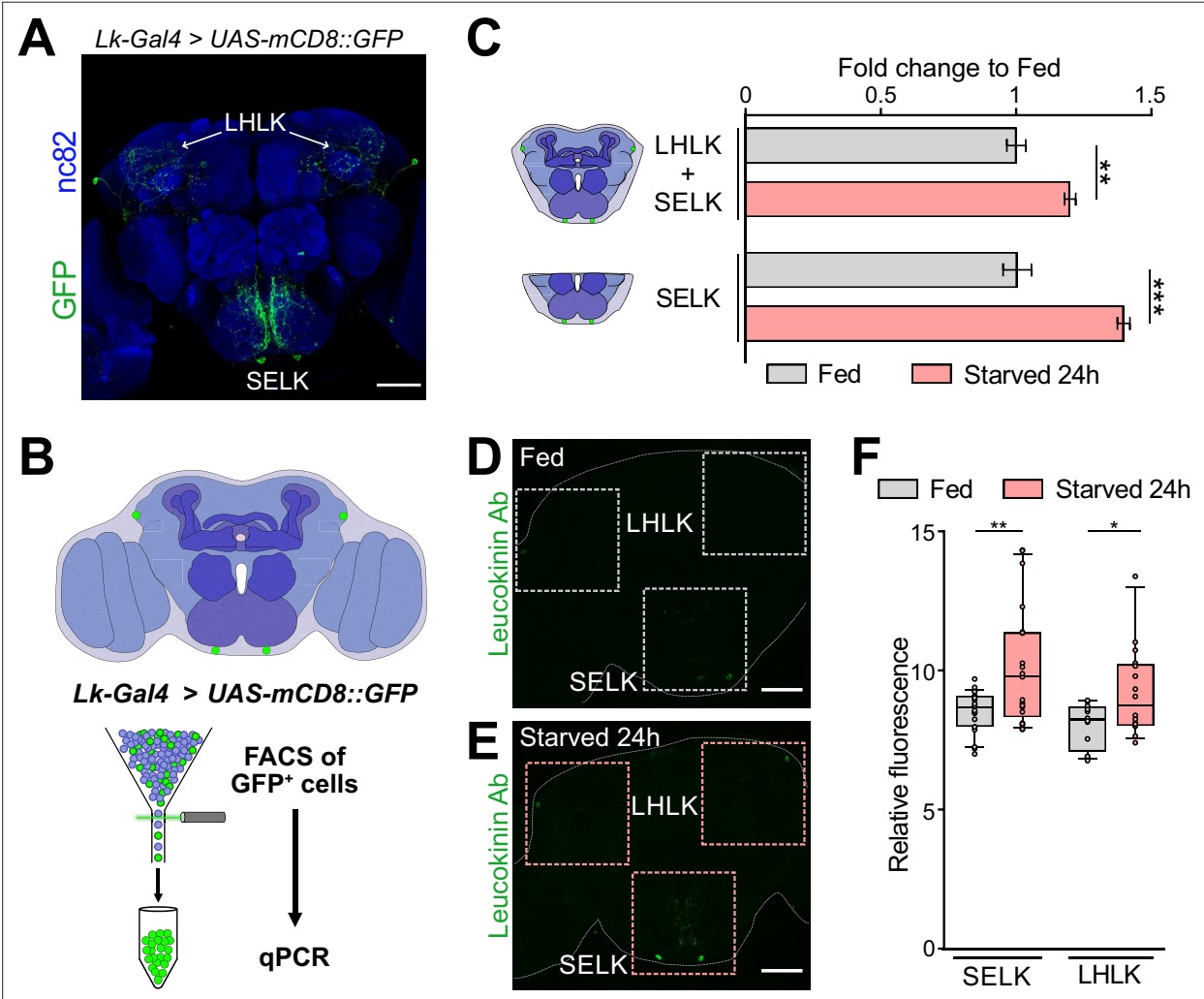

**Figure 3.** *Leucokinin* expression is increased in starvation. (**A**) Immunofluorescence with anti-GFP (green) and anti-nc82 (blue) on a whole-mount brain of a *Lk-Gal4 >UAS-mCD8::GFP*. (**B**) Scheme showing the approach used to select the Lk GFP⁺ neurons by Fluorescent Activated Cell Sorting (FACS). (**C**) qPCR results for Lk neurons labeled with GFP and sorted by FACS. The central brain includes four Lk neurons, while the SEZ region includes only the two neurons located in this region. (**D**) Intensity fluorescence quantification in whole *Oregon-R* brains stained with anti-Lk antibody in fed and 24 hr starved flies. SELK Fed = 20 brains; SELK Starved = 21 brains; LHLK Fed = 12 brains; LHLK Starved = 16 brains. Statistical test: Wilcoxon rank sum test. *p<0.05, **p<0.01, ****p<0.0001. Scale bar = 50 µm.

*2016*; *Murphy et al., 2016*; *Yurgel et al., 2019*). Functional imaging revealed that LHLK neurons, not SELK neurons, were required to suppress sleeping during starvation (*Yurgel et al., 2019*). It has been suggested that SELK neurons may regulate feeding as possible synaptic partners to the premotor feeding program. However, it is still not clear the role of SELK neurons in feeding behavior (*Nässel, 2021*; *Zandawala et al., 2018*).

Due to the dissection method used for our RNAseq experiment (*Figure 1A*), we primarily captured SELK neurons rather than LHLK neurons. To confirm that starvation induces changes in *Lk* gene expression, we performed qPCR on brains from fed and starved *Drosophila*. Given that Lk neurons constitute 4 out of approximately 200,000 neurons in the *Drosophila* brain (*Raji and Potter, 2021*), we enriched our samples by expressing GFP in Lk neurons using the driver *Lk-Gal4* (*Asahina et al., 2014*) combined with *UAS-mCD8::GFP* transgene and FACS of the GFP⁺ cells. We conducted qPCR analysis on the whole CB (including both LHLK and SELK neurons) as well as on the SEZ alone (containing only the 2 SELK neurons) in fed and starved conditions (*Figure 3A and B*). Our results showed that starvation increased the expression level of *Lk* mRNA in the CB and also in the SELK neurons, indicating that *Lk* expression is indeed affected by the animal's starvation state (*Figure 3C*). To confirm that the increased mRNA expression translates into higher protein levels, we performed immunohistochemistry using an Lk antibody in fed and 24 hr starved wild-type flies. We quantified the fluorescence in the LHLK and SELK neurons and found that *Lk* expression was elevated under starvation conditions (*Figure 3D, E and F*), further validating our RNAseq results. Together, these three lines of evidence indicate that SELK neurons increase the expression of the Lk neuropeptide in 24 hr starved flies.

## Leucokinin neurons are gustatory second-order neurons to GRNs

Our RNAseq data showed that the two SELK neurons receive almost exclusive input from bitter, but not sweet, GRNs and that the metabolic state of the fly modulates the expression of *Lk*. To confirm that Lk neurons are indeed postsynaptic partners of bitter GRNs, we used an Lk antibody (*de Haro et al.,*

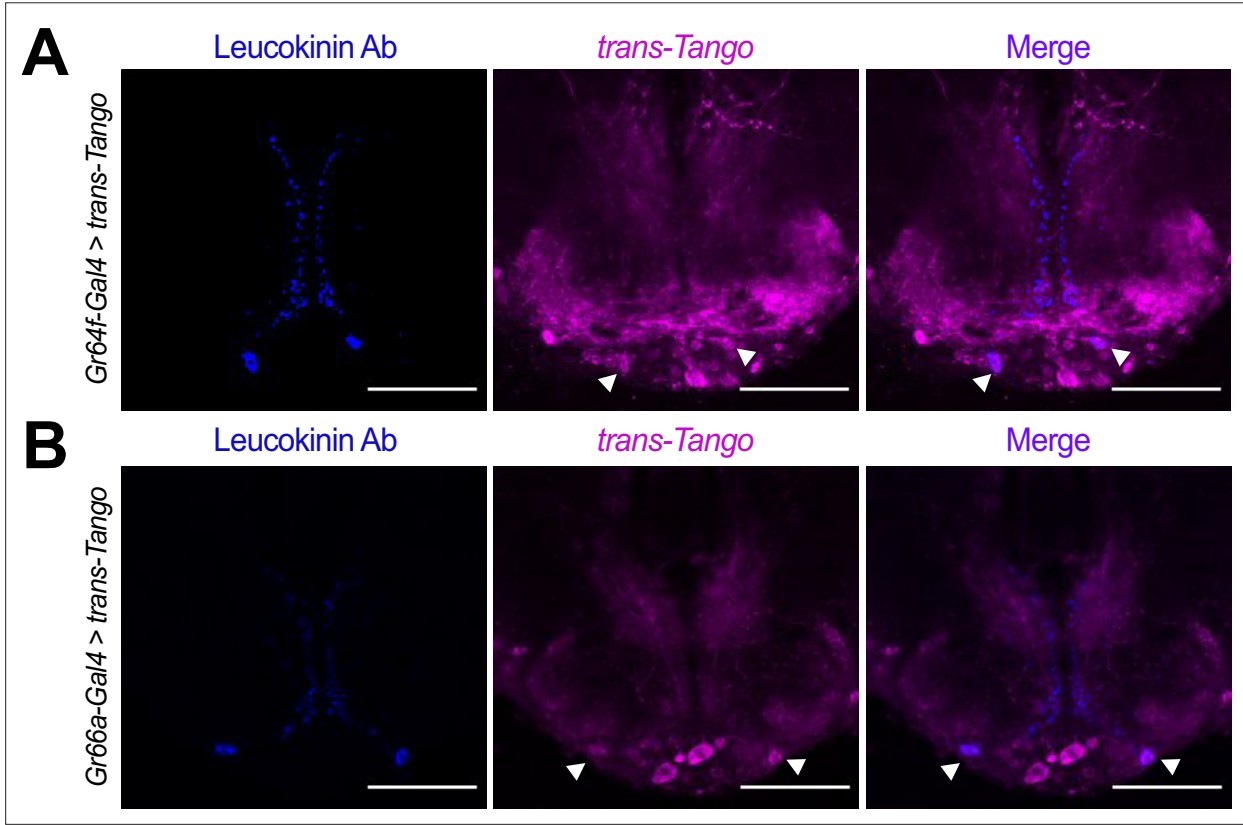

**Figure 4.** Lk neurons co-localize with the *trans*-Tango signal. Immunofluorescence with anti-Leucokinin (blue) and anti-RFP (magenta) on whole-mount brains of (**A**) *Gr64f-Gal4>trans-Tango*, and (**B**) *Gr66a-Gal4>trans-Tango*. Images correspond to the adult Subesophageal Zone (SEZ). Arrowheads point to SELK neurons. Scale bar = 50 μm.

*2010*) to test the co-localization of Lk protein with the cells labeled in *GRNs-Gal4 >trans-Tango* flies. These experiments demonstrated that Lk signal colocalizes with trans-Tango in *Gr66a-Gal4>trans-Tango* flies, confirming our expectations based on the RNAseq data (*Figure 4A*). Unexpectedly, however, we also found co-localization of the Lk antibody signal in *Gr64f-Gal4>trans-Tango* flies, suggesting that SELK neurons do not receive exclusive input from bitter neurons but also from sweet-sensing neurons expressing the Gr64f receptor (*Figure 4B*).

To validate the synaptic connectivity between Gr64f[GRNs] and Gr66a[GRNs] with SELK neurons, we next used the GRASP (GFP Reconstruction Across Synaptic Partners) technique (*Feinberg et al., 2008*). GRASP can reveal the synaptic connectivity between two neurons, through the expression of half of the GFP protein in each candidate neuron. The reconstruction of the entire GFP protein in the synaptic cleft can be detected by a specific GFP antibody. We expressed half of the GFP (GFP *Scott, 2018*) protein in the bitter GRNs using *Gr66a-LexA* (*Thistle et al., 2012*) and sweet GRNs using *Gr64f-LexA* (*Fujii et al., 2015*) transgenes. The other half of the GFP (GFP *Chu et al., 2014*; *Inagaki et al., 2014*; *LeDue et al., 2016*; *Yarmolinsky et al., 2016*; *Vosshall and Stocker, 2007*; *Montell, 2021*; *Dahanukar et al., 2007*; *Fujii et al., 2015*; *Dunipace et al., 2001*; *Wang et al., 2004*) protein was expressed in Lk neurons using the *Lk-Gal4* driver. GRASP results confirmed the synaptic connection between bitter GRNs and sweet GRN with SELK neurons in the SEZ (*Figure 5A and C*), further confirming our previous *trans*-Tango result.

The GRASP technique can produce false positives, as a positive signal may result from the proximity of the neurons of interest. We decided to complement those results using active-GRASP (*Macpherson et al., 2015*). In this modified version, the presynaptic half of GFP (GFP *Chu et al., 2014*; *Inagaki et al., 2014*; *LeDue et al., 2016*; *Yarmolinsky et al., 2016*; *Vosshall and Stocker, 2007*; *Montell, 2021*; *Dahanukar et al., 2007*; *Fujii et al., 2015*; *Dunipace et al., 2001*; *Wang et al., 2004*) is fused to synaptobrevin. This fusion allows the GFP to be present in the membrane of the presynaptic neuron within the synaptic cleft when the presynaptic neuron is stimulated. In our experiments, we fed the flies with sucrose to activate sweet GRNs and caffeine to activate bitter GRNs. A positive signal in active-GRASP indicates synaptic connectivity between the candidate neurons and active communication between them. Our results with active-GRASP supported the findings from GRASP. Stimulation with sucrose resulted in a clear positive GRASP signal in the *Gr64f-lexA, Lk-Gal4 >active-GRASP* flies, whereas stimulation with water did not produce a positive result (*Figure 5B*). Similarly, caffeine stimulation produced a strong positive GRASP signal in the *Gr66a-LexA, Lk-Gal4 >active-GRASP* flies, that was absent in the control group (water) (*Figure 5D*) (see Materials and Methods for a full description of the stimulation procedure). Our GRASP and active-GRASP results indicate that SELK neurons receive active input from sweet Gr64f[GRNs] and bitter Gr66a[GRNs]. Finally, we used BacTrace (*Cachero et al., 2020*) to confirm that the connectivity between Gr64f[GRNs] and Gr66a[GRNs] neurons with SELK neurons is real. This technique works similarly to *trans*-Tango, but while *trans*-Tango labels postsynaptic neurons, BacTrace labels presynaptic partners. For our experiments, we used *Gr66a-LexA* and *Gr64f-LexA* as candidate presynaptic partners. Consistent with the GRASP and active-GRASP results, our BacTrace experiment indicated that sweet and bitter GRNs are presynaptic to SELK neurons (*Figure 5E*) further supporting a model in which Lk neurons are G2Ns receiving input from Gr64f[GRNs] and Gr66a[GRNs]. To our best knowledge, this is the first time a G2N has been identified as collecting information from two populations of GRNs that transduce sensory information of different valence, sweet-attractive and bitter-repulsive.

## Connectivity of SELK neurons within the *Drosophila* brain

Using molecular techniques, we have demonstrated that SELK neurons are G2Ns for both bitter Gr66a[GRNs] and sweet Gr64f[GRNs]. We used the recently published Full Adult Fly Brain (FAFB) *Drosophila* connectome (*Zheng et al., 2018*) to identify the SELK neurons and further analyze their connectivity. As SELK neurons receive simultaneous input from sweet and bitter GRNs, we took advantage of recent research that identified groups of bitter and sweet-sensing GRNs and their projections to the SEZ in the *Drosophila* brain connectome (*Figure 6—figure supplement 1A*; *Engert et al., 2022*). Using this dataset, we identify all postsynaptic partners to the Gr64f[GRNs] and Gr66a[GRNs] characterized, without limiting the number of synaptic contacts among neurons. We noted 12594 postsynaptic hits for Gr66a[GRNs] and 36790 hits for Gr64f[GRNs]. By intersecting both datasets, we obtained 333 downstream segment IDs as possible G2Ns receiving inputs from both GRNs. We refined our

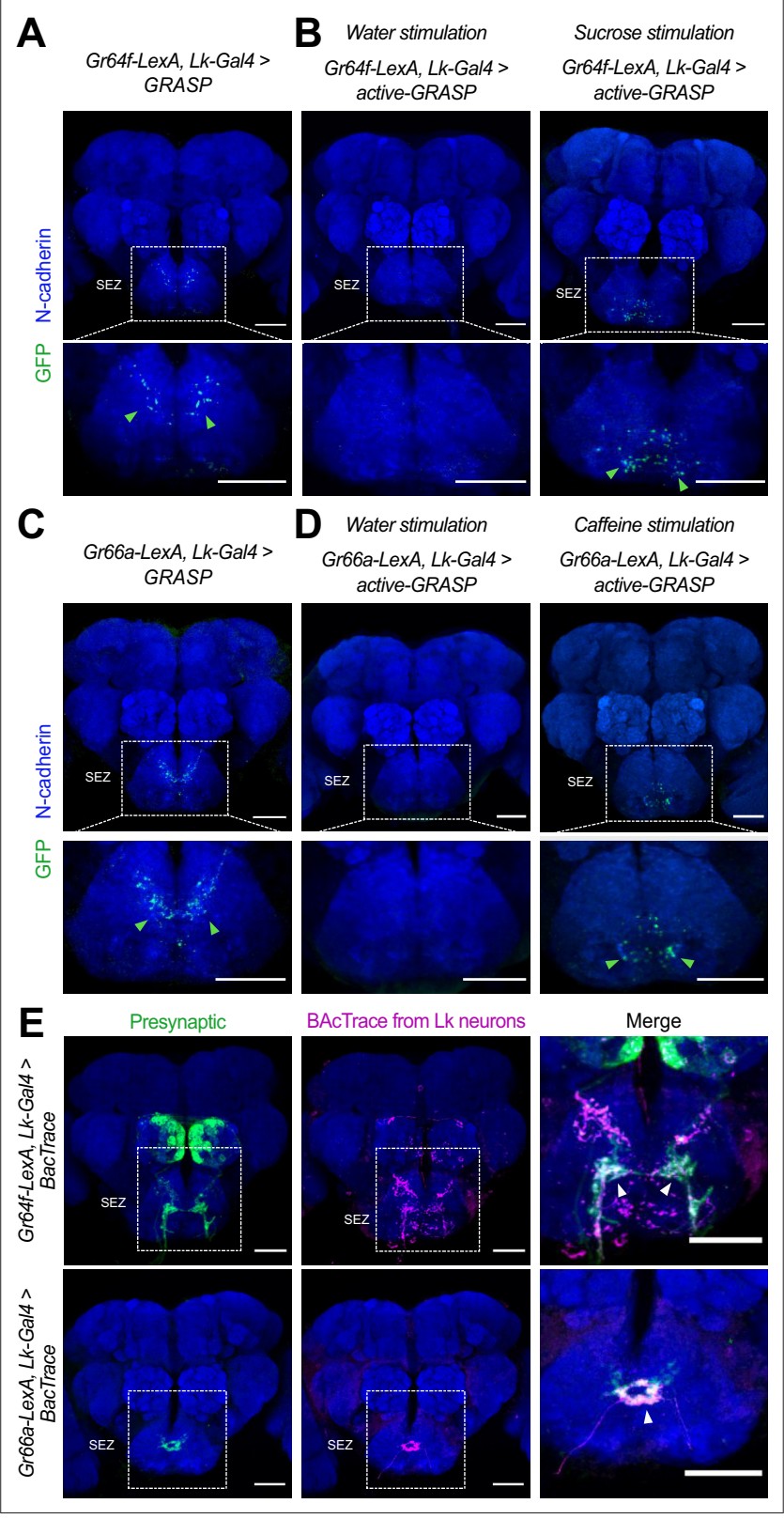

**Figure 5.** Lk neurons are synaptically connected to Gr64f[GRNs] and Gr66a[GRNs]. (**A**) GRASP: Immunofluorescence with anti-GFP (green) and anti-N-cadherin (blue) on whole-mount brains of Gr64f-LexA>LexAop-CD4spGFP (*Scott, 2018*) and Lk-Gal4 >UAS-CD4spGFP (*Chu et al., 2014*; *Inagaki et al., 2014*; *LeDue et al., 2016*; *Yarmolinsky et al., 2016*; *Vosshall and Stocker, 2007*; *Dahanukar et al., 2007*; *Fujii et al., 2015*; *Dunipace et al., 2001*;

*Figure 5 continued on next page*

*Figure 5 continued*

*Wang et al., 2004* flies). (**B**) Active-GRASP: Immunofluorescence with anti-GFP (green) and anti-N-cadherin (blue) on whole-mount brains of Gr64f-LexA>LexAop-nSybspGFP (*Chu et al., 2014*; *Inagaki et al., 2014*; *LeDue et al., 2016*; *Yarmolinsky et al., 2016*; *Vosshall and Stocker, 2007*; *Dahanukar et al., 2007*; *Fujii et al., 2015*; *Dunipace et al., 2001*; *Wang et al., 2004*) and Lk-Gal4 >UAS-CD4spGFP (*Scott, 2018*) stimulated with water (left column) and sucrose (right column). (**C**) GRASP: Immunofluorescence with anti-GFP (green) and anti-N-cadherin (blue) on whole-mount brains of Gr66a-LexA>LexAop-CD4spGFP (*Scott, 2018*) and Lk-Gal4 >UAS-CD4spGFP (*Chu et al., 2014*; *Inagaki et al., 2014*; *LeDue et al., 2016*; *Yarmolinsky et al., 2016*; *Vosshall and Stocker, 2007*; *Dahanukar et al., 2007*; *Fujii et al., 2015*; *Dunipace et al., 2001*; *Wang et al., 2004*). (**D**) Active-GRASP: Immunofluorescence with anti-GFP (green) and anti-N-cadherin (blue) on whole-mount brains of Gr66a-LexA>LexAop-nSybspGFP (*Chu et al., 2014*; *Inagaki et al., 2014*; *LeDue et al., 2016*; *Yarmolinsky et al., 2016*; *Vosshall and Stocker, 2007*; *Dahanukar et al., 2007*; *Fujii et al., 2015*; *Dunipace et al., 2001*; *Wang et al., 2004*) and Lk-Gal4 >UAS-CD4spGFP (*Scott, 2018*) stimulated with water (left column) and caffeine (right column). (**E**) Immunofluorescence with anti-GFP (green), anti-RFP (magenta), and anti-nc82 (blue) on whole-mount brains of Gr64f-LexA, Lk-Gal4 >BacTrace (top row) and Gr66a-LexA, Lk-Gal4 >BacTrace (bottom row). Arrowheads point to the BacTrace positive signal. Scale bar = 50 µm.

list to 17 candidates through visual inspection, focusing on neurons with projections within the SEZ (*Figure 6—figure supplement 1B*). To identify the SELK neuron among our candidates, we used the CMTK plugin in ImageJ to register the immunohistochemistry images of the *Lk-Gal4 >UAS-CD8::GFP* brains (*Figure 3A*) to a *Drosophila* reference brain (JRC2018U). Further, we skeletonized the images with ImageJ and uploaded them to the Flywire Gateway platform to compare the structure of the aligned SELK neurons with our candidates (*Figure 6—figure supplement 1C*). Via comparison with our candidates, we identified one candidate ID 720575940630808827 (GNG.685) (*Figure 6—figure supplement 1D*), which shared similar morphologies and projections to the right SELK neuron. The GNG.685 has a mirror twin, GNG.595, in the SEZ, and both are annotated as DNg68 in the Flywire database (*Figure 6A*). These candidates are classified as descending neurons, characteristic of known SELK neurons (*Nässel, 2021*) and possibly also expressing the neurotransmitter acetylcholine. We consider the neurons DNg68 are the strongest candidates for representing SELK neurons (see Materials and methods for a more detailed description).

To further analyze the connectivity of these neurons, we looked for the pre- and postsynaptic organization of SELK neurons. We only considered those synaptic partners with ≥10 synaptic connections to limit possible false positive synaptic connections. Most of the postsynaptic partners of SELK neurons (57) localized in the SEZ, as revealed by the BrainCircuits analysis (*Gerhard et al., 2023*; *Figure 6B*), with significant projections to the pars intercerebralis. This region contains Insulin-Producing Cells (IPCs) that project to the tritocerebrum in the SEZ, where a significant concentration of fibers can be appreciated in the *trans*-Tango immunohistochemistry (*Figure 6C*). This data supports previous results showing that IPCs express the *Lk receptor* (*Lkr*) and that they may be modulated by Lk-expressing neurons (*Zandawala et al., 2018*; *Yurgel et al., 2019*). We found five bitter (GRN R1#4, GRN R1#5, GRN R1#12, GRN R2#1, and GRN R2#2) and three sweet (GRN R4#17, GRN R5#12, and GRN R5#14) GRN as presynaptic candidates to GNG.685 (right SELK neuron), and only one bitter (GRN R1#1) and one sweet (GRN R4#11) GRN as presynaptic candidates to GNG.595 (left SELK neuron) (IDs correspond to GRNs described in *Engert et al., 2022*; Virtual Fly Brain). That so few presynaptic GRNs were found indicates that SELK proofreading is still required for a much more precise assembly and annotation of the FAFB dataset. We identified a total of 101 presynaptic partners using BrainCircuits with a complex distribution over the brain (*Figure 6D*), though most of the connectivity was found within the SEZ. Additionally, we used *retro*-Tango (*Sorkaç, 2023*), a retrograde trans-synaptic technique that employs similar trans-synaptic mechanisms as *trans*-Tango but in the retrograde direction. This experiment showed that SELK neurons receive most of the presynaptic input from neurons located in the SEZ (*Figure 6E*), consistent with the data derived from the FAFB analysis. Encouragingly, we observed a significant concentration of synapse at the tritocerebrum, similar to the *trans*-Tango signal.

## Leucokinin modulates feeding behavior

We have shown that Lk neurons receive direct input from GRNs for bitter (Gr66a[GRNs]) and sweet (Gr64f[GRNs]), with Lk neuropeptide expression influenced by the animal's starvation level. Given that Gr64f[GRNs] facilitates feeding while Gr66a[GRNs] inhibit it, we sought to understand how Lk neurons

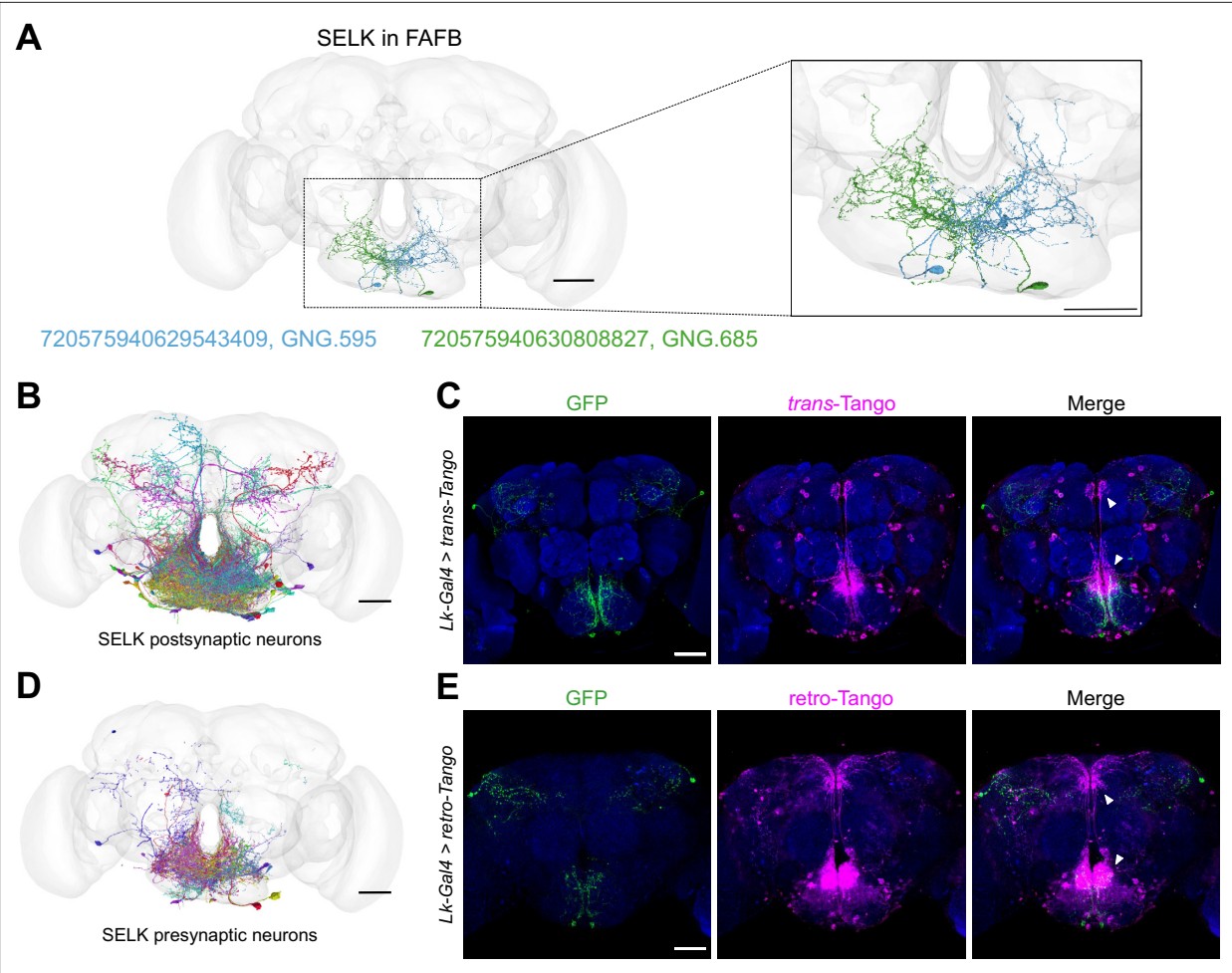

**Figure 6.** Lk neurons are pre- and postsynaptic to a significant number of neurons. (**A**) The anatomy of the SELK neurons (Right in green and Left in blue) was reconstructed using the FAFB brainmesh template in Flywire. IDs from Flywire are indicated: 720575940629543409 ID for the Left SELK in blue (GNG.595/DNg68(L)) and 720575940630808827 for the Right SELK in green (GNG.685/DNg68(R)). Zoom in to the Subesophageal Zone (SEZ) from (**A**), showing the details of the neuronal arborizations. (**B**) SELK postsynaptic candidate neurons reconstructed in the FAFB (Full Adult Fly Brain) brainmesh template in Flywire. Only postsynaptic candidate neurons with ≥10 synaptic points with SELK neurons are shown (57 postsynaptic candidate neurons). (**C**) Immunofluorescence with anti-GFP (green), anti-RFP (magenta), and anti-nc82 (blue) on a whole-mount brain of a *Lk-Gal4 >trans-Tango* flies. (**D**) SELK presynaptic candidate neurons reconstructed in the FAFB brain mesh template in Flywire. Only presynaptic candidate neurons with ≥10 synaptic points with SELK neurons are shown (101 presynaptic candidate neurons). (**E**) Immunofluorescence with anti-GFP (green), anti-RFP (magenta), and anti-nc82 (blue) on a whole-mount brain of a *Lk-Gal4 >retro-Tango* fly. Scale bar: 50 μm.

The online version of this article includes the following figure supplement(s) for figure 6:

**Figure supplement 1.** Sequential methodology to identify a strong SELK candidate neuron in the FAFB connectome.

integrate gustatory and metabolic signals to modulate feeding behavior. Previous studies indicated that silencing Lk neurons impaired flies' responses to sucrose in a Proboscis Extension Response (PER) experiment (*Zandawala et al., 2018*). We silenced all Lk neurons in *Drosophila* using the *Lk-Gal4* driver to express the tetanus toxin (TNT) using the *UAS-TNT* driver alongside *UAS-TNT^{imp}* controls (*Sweeney et al., 1995*), then assessed their response in a PER assay (*Figure 7A*). The light chain of the tetanus toxin expressed with the *UAS-TNT* transgene, blocks all chemical transmission by cleaving the presynaptic protein nSyb. Unexpectedly, silencing Lk neurons did not change the gustatory response to varying sucrose concentrations in 24 hr starved flies (*Figure 7B*). Discrepancies with previous findings may stem from differences in handling and stimuli methodologies (*Zandawala et al., 2018*).

We hypothesized that Lk neurons might be important for processing sweet and bitter information. To test their role in the PER response to sweet food laced with bitter compounds, we combined 50 mM sucrose with increasing caffeine levels. It is known that *Drosophila* PER to a certain sucrose

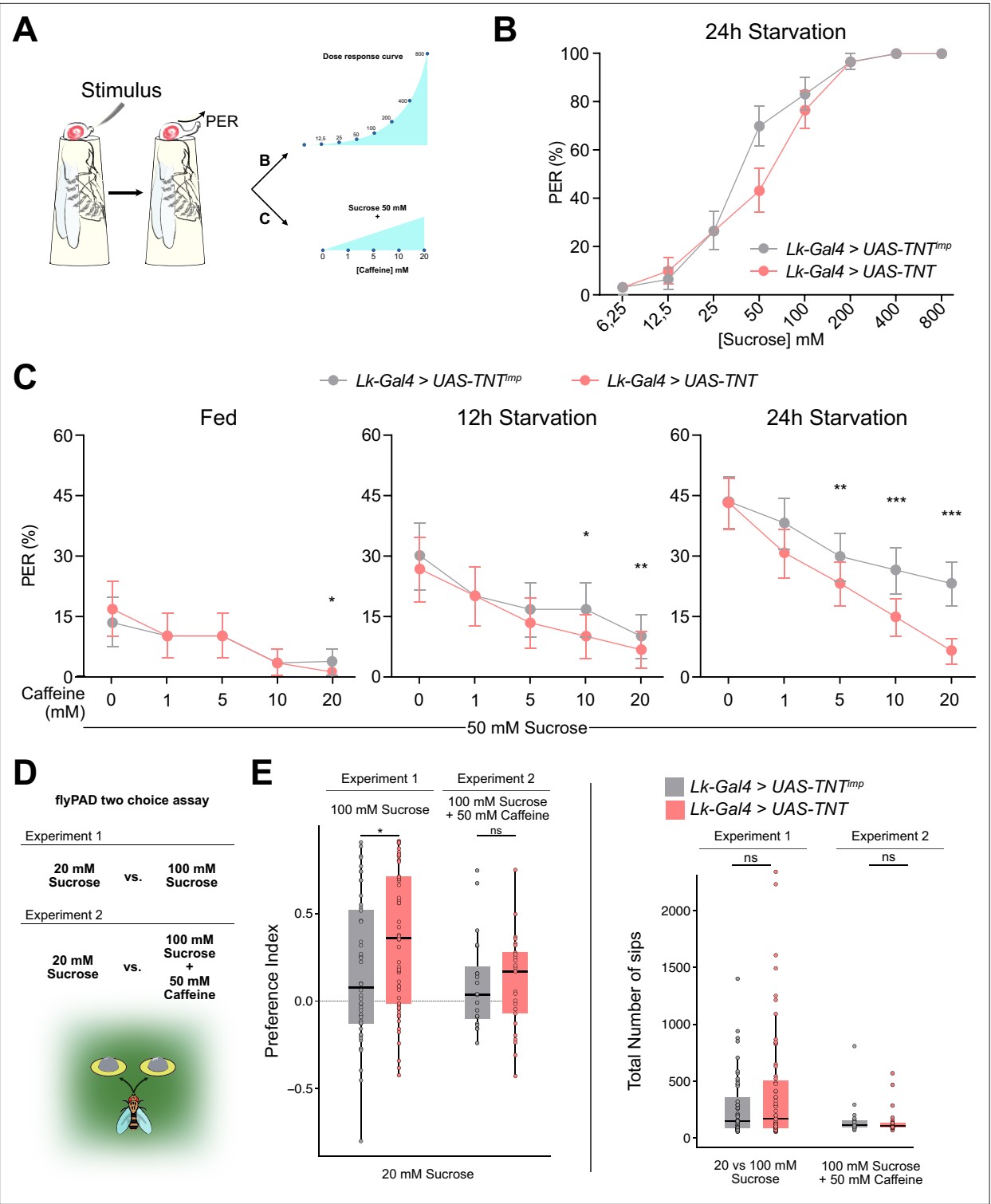

**Figure 7.** Leucokinin neurons integrate sweet and bitter gustatory information. (**A**) Scheme depicting the experimental procedure for the Proboscis Extension Reflex (PER). Flies are given a series of increasing concentrations of sucrose (6.25, 12.5, 25, 50, 100, 200, 400, and 800 mM) for Experiment 1; and a fixed concentration of sucrose (50 mM) mixed with increasing concentrations of caffeine (1, 5, 10, and 20 mM) for Experiment 2. All flies starved 24 hr. (**B**) PER dose response curve of *Lk-Gal4 >UAS-TNT^imp* (n=30) and *Lk-Gal4 >UAS* TNT (n=30) flies for Experiment 1. (**C**) PER of *Lk-Gal4 >UAS-TNT^imp* (n=30) and *Lk-Gal4 >UAS* TNT (n=30) flies for Experiment 2. (**D**) Scheme depicting the experimental procedure for flyPAD (Adapted from *Itskov et al., 2014*). Experiment 3: 20 mM Sucrose vs. 100 mM Sucrose and Experiment 4: 20 mM Sucrose vs. 100 mM Sucrose +50 mM Caffeine. All flies starved for 24 hr. (**E**) Preference Index (Left) for the highest concentration of sucrose (100 mM) with or without 50 mM caffeine and total number of sips

*Figure 7 continued on next page*

*Figure 7 continued*

(Right) for Experiments 3 and 4. Experiment 3: *Lk-Gal4 >UAS-TNT^imp* (n=48); *Lk-Gal4 >UAS* TNT (n=59); Experiment 4: *Lk-Gal4 >UAS-TNT^imp* (n=20); *Lk-Gal4 >UAS* TNT (n=25). Only flies that performed a minimum of 25 sips per fly and two bouts were considered for the analysis. ns: non-significant, *p<0.05. PER analysis: Logistic regression model with binomial distribution; Error bars represent the standard error of the proportion. flyPAD statistics Wilcoxon Rank Sum Test.

The online version of this article includes the following figure supplement(s) for figure 7:

**Figure supplement 1.** Leucokinin neurons are essential to discriminate sweet and bitter stimuli.

concentration decreases proportionally to the amount of a bitter compound (i.e. caffeine) it contains (*Inagaki et al., 2014*; *Figure 7A*, Experiment 2). The PER experiments included fed, 12 hr, and 24 hr starved flies (*Figure 7C*). Fed flies showed a minimal response to 50 mM sucrose, with caffeine reducing this response consistently across both control and experimental groups. Twelve-hour-starved flies exhibited stronger reactions to sucrose than fed flies, with caffeine diminishing PER in a concentration-dependent manner. Twenty-four-hour starved flies showed larger PER values that decreased proportionally to the concentration of caffeine. However, this decrease was significantly larger when Lk neurons were silenced using TNT, indicating that Lk neurons are needed to tolerate and feed on food containing bitter, and potentially toxic, compounds. To restrict *Lk-Gal4* transgene expression, we employed the *tsh-Gal80* (*Clyne and Miesenböck, 2008*) construct, which is expressed in the Ventral Nerve Cord (VNC) to restrict Gal4 expression in this region. Restricting TNT expression to SELK and LHLK neurons, did not alter the previously observed PER responses to sweet and bitter compounds (*Figure 7—figure supplement 1A*). To investigate the role of Leucokinin further, we employed an RNAi line designed to knock down Leucokinin production in Lk-expressing neurons. The results presented in *Figure 7—figure supplement 1B* demonstrate that knocking down Leucokinin significantly diminishes the flies' tolerance to caffeine in sweet food, indicating that this neuropeptide might be involved in the integration of sweet and bitter signals in Lk neurons.

Finally, we examined whether Lk neurons influence sustained feeding using the flyPAD (*Itskov et al., 2014*) to monitor feeding behavior over 1 hr. We designed two feeding assays (*Figure 7D*). In Experiment 1, flies chose between 20 mM and 100 mM sucrose food. Typically, flies prefer higher sucrose concentrations, but if the higher concentration is mixed with a bitter compound (e.g. caffeine), acceptance of the lower concentration correlates with caffeine levels. Flies with silenced Lk neurons slightly preferred the higher sucrose concentration to controls (*Figure 7E*), yet caffeine addition did not affect their choices. The ingestion amounts for each fly remained comparable in both scenarios. Our findings indicate that Lk neurons are needed to tolerate and initiate feeding of food laced with bitter compounds; however, their role in sustained feeding remains unclear.

## Discussion

We have demonstrated that G2Ns, which transduce opposing behaviors such as sweet attraction and bitter repulsion, exhibit molecular differences and undergo transcriptomic changes upon starvation. Molecular, connectomic, and behavioral experiments have identified SELK neurons as particularly dynamic G2Ns when under starved conditions. Surprisingly, we discovered that these neurons simultaneously collect information from sweet-sensing Gr64f^GRNs and bitter-sensing Gr66a^GRNs. Additionally, SELK neurons respond to starvation by increasing the neuropeptide Leucokinin (Lk) expression. These findings reveal that Leucokinin neurons play a novel role in modulating feeding initiation during starvation when flies are presented with sweet food laced with bitter compounds (*Figure 8*).

Our RNAseq data analysis revealed that SELK neurons only received input from Gr66a^GRNs. However, further analysis using immunohistochemistry (*GRN-Gal4 >trans-Tango*+Lk immunohistochemistry) combined with GRASP, active-GRASP, and BacTrace, demonstrated that SELK neurons are G2Ns to Gr64f^GRNs and Gr66a^GRNs. To our knowledge, this is the first demonstration that a pair of G2Ns is broadly tuned to both bitter and sweet gustatory inputs while also integrating metabolic information. *trans*-Tango and its variants have been successfully used to study connectivity and functionality in various neuronal circuits, including gustatory, olfactory, visual systems, mushroom body output neurons or clock neurons (*Talay et al., 2017*; *Snell et al., 2022*; *Kiral et al., 2021*; *Scaplen et al., 2021*; *Ehrlich et al., 2024*). Recent analysis of the *retro*-Tango tool showed that a minimum number of synaptic connections between neurons is needed to label the postsynaptic neurons properly. It is

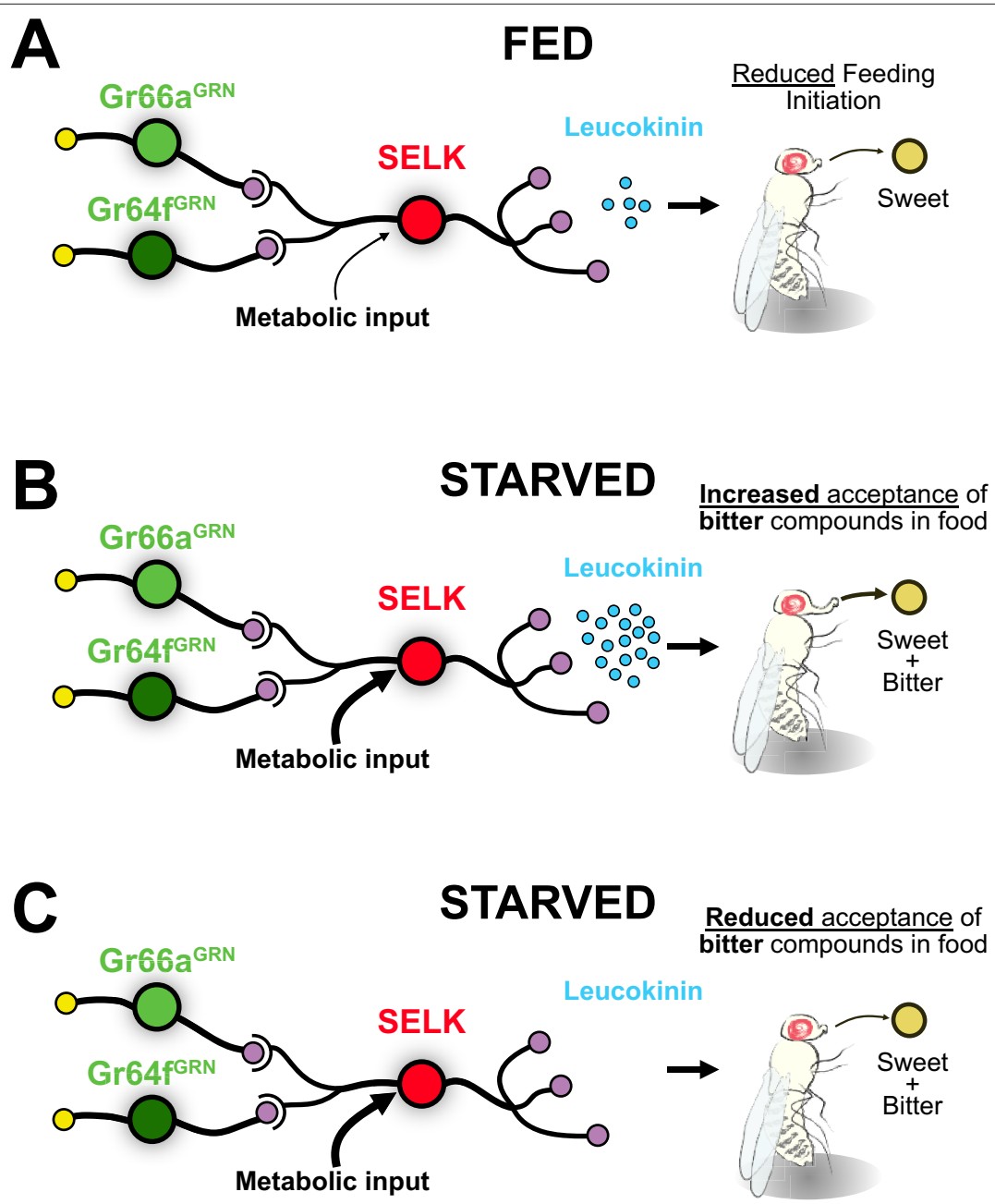

**Figure 8.** Model for the integration of gustatory information by SELK neurons. (**A**) SELK neurons receive direct input from Gr64f^GRNs (sweet), Gr66a^GRNs (bitter), and metabolic information. When flies are fed, the expression of the neuropeptide Leucokinin is reduced. The probability of a fed fly initiating feeding is reduced when facing any food. (**B**) SELK neurons in starved flies express large amounts of Leucokinin. In this situation, flies are more prone to accept food containing bitter compounds. (**C**) Starved flies with silenced SELK neurons are less prone to accept sweet foods laced with bitter compounds.

possible that Gr64f^GRNs makes fewer synaptic contacts with SELK neurons than Gr66a^GRNs, decreasing the probability of being labeled by *trans*-Tango. Combined with our stringent FACS protocol, could explain why we were unable to capture SELK neurons as Gr64f^G2Ns with FACS.

Current understanding suggests that tastant quality is mediated by labeled lines, where gustatory receptor neurons are segregated to respond to specific taste qualities (*Yarmolinsky et al., 2009*). This segregation is thought to persist further upstream in the circuit, as observed in mice, where taste cells respond to specific tastants, and this information remains segregated at the geniculate ganglion (*Barretto et al., 2015*), gustatory cortex (*Chen et al., 2011*), and the amygdala (*Wang et al.,*

2018). Similarly, in *Drosophila* GRNs that detect bitter and sweet compounds project their axons to the SEZ in a non-overlapping fashion (*Wang et al., 2004*). Whole-brain in vivo calcium imaging revealed that most SEZ interneurons respond to single tastants (*Harris et al., 2015*). However, the same study identified neurons that may respond to multiple tastants, like bitter and sweet, though specific neurons and their circuit positions were not detailed. Other SEZ neurons have been shown to integrate sweet and bitter inputs, such as the E49 motor neuron, which is stimulated by sweet input (Gr5a) and inhibited by bitter input (Gr66a) (*Gordon and Scott, 2009*), but these are not G2Ns. We attempted in vivo calcium imaging to further study the gustatory integration in those neurons, but we were unsuccessful due to the challenging location of SELK somas in the medio-ventral region of the SEZ, which is obscured by the proboscis even when fully extended. Additionally, variability in labeling using *UAS-GCaMP7s*, *UAS-GCaMP7m*, and *UAS-GCaMP6f* lines hampered the proper imaging of these neurons. Our behavior experiments show that Leucokinin neurons (most likely SELK neurons) are early integrators of gustatory and metabolic information.

In summary, our study highlights the complex integration of gustatory and metabolic information by Leucokinin neurons, providing new insights into the neural mechanisms underlying feeding behavior in *Drosophila*. This work advances our understanding of how sensory and metabolic cues are integrated into the brain to regulate vital behaviors such as feeding.

## Materials and methods

**Key resources table**

| Reagent type (species) or resource | Designation | Source or reference | Identifiers | Additional information |
|---|---|---|---|---|
| Strain (*D. melanogaster*) | *Oregon-R* | Kindly donated by Prof. Richard Benton | NA | |
| Strain (*D. melanogaster*) | *w1118* | Kindly donated by Prof. Francisco Tejedor | NA | |
| Strain (*D. melanogaster*) | *Gr64f-Gal4* | Kindly donated by Prof. Richard Benton | NA | |
| Genetic reagent (*D. melanogaster*) | *Gr64f-LexA* | Kindly donated by Prof. Richard Benton | NA | |
| Genetic reagent (*D. melanogaster*) | *Gr66a-Gal4* | Bloomington *Drosophila* Stock Center | BDSC 57670 | |
| Genetic reagent (*D. melanogaster*) | *Gr66a-LexA; LexAop-rCD2::GFP* | Kindly donated by Prof. Richard Benton | NA | |
| Genetic reagent (*D. melanogaster*) | *Lk-Gal4 (2nd Chr)* | Bloomington *Drosophila* Stock Center | BDSC 51993 | |
| Genetic reagent (*D. melanogaster*) | *Lk-Gal4 (X Chr) used only in the tsh-Gal80 experiment* | Bloomington *Drosophila* Stock Center | BDSC 51992 | |
| Genetic reagent (*D. melanogaster*) | *retro-Tango (QUAS-mtdTomato; retro-Tango; UAS-retro-Tango-GFP)* | Bloomington *Drosophila* Stock Center | BDSC 99661 | |
| Genetic reagent (*D. melanogaster*) | *UAS-mCD8::GFP* | Bloomington *Drosophila* Stock Center | BDSC 32186 | |
| Genetic reagent (*D. melanogaster*) | *trans-Tango (UAS-myrGFP, QUAS-mtdTomato; trans-Tango)* | Bloomington *Drosophila* Stock Center | BDSC 77124 | |
| Genetic reagent (*D. melanogaster*) | *UAS-TNT* | Bloomington *Drosophila* Stock Center | BDSC 28837 | |
| Genetic reagent (*D. melanogaster*) | *UAS-TNTimp* | Bloomington *Drosophila* Stock Center | BDSC 28839 | |
| Genetic reagent (*D. melanogaster*) | *UAS-RNAiLk* | Bloomington *Drosophila* Stock Center | BDSC 25798 | |
| Genetic reagent (*D. melanogaster*) | GRASP (*UAS-CD4-spGFP1-10, lexAop-CD4-spGFP11*) | Bloomington *Drosophila* Stock Center | BDSC 58755 | |

*Continued on next page*

*Continued*

| Reagent type (species) or resource | Designation | Source or reference | Identifiers | Additional information |
|---|---|---|---|---|
| Genetic reagent (*D. melanogaster*) | Active GRASP (*lexAop-nSyb-spGFP1-10, UAS-CD4-spGFP11; MKRS/TM6B*) | Bloomington *Drosophila* Stock Center | BDSC 64315 | |
| Genetic reagent (*D. melanogaster*) | BAcTrace (*LexAop-GFP, LexAop-QF; UAS-B3RT, QUAS-mtdTomato*) | Bloomington *Drosophila* Stock Center | BDSC 90826 | |
| Genetic reagent (*D. melanogaster*) | *tsh-Gal80* | Kindly donated by Prof. María Domínguez | | |
| Antibody | GFP (Green Fluorescent Protein) | Abcam | ab13970 | 1:500 |
| Antibody | RFP (Red Fluorescent Protein) | Abcam | ab62341 | 1:500 |
| Antibody | nc82 Bruchpilot | DSHB | | 1:50 |
| Antibody | GFP reconstructed in GRASP | Sigma-Aldrich | G6539 | 1:500 |
| Antibody | Leucokinin | Kindly donated by Prof. Pilar Herrero | | 1:100 |
| Antibody | *Cadherin DN-Extracellular Domain* | Jackson Inmunoresearch | 112-165-003 | 1:500 |
| Antibody | Anti-Chicken Alexa 488 | Abcam | ab150169 | 1:250 |
| Antibody | Anti-Rabbit Cy3 | Jackson Inmunoresearch | 111-165-144 | 1:250 |
| Antibody | Anti-Mouse Cy5 | Jackson Inmunoresearch | 115-175-146 | 1:250 |
| Antibody | Anti-Mouse Cy3 | Jackson Inmunoresearch | 115-165-166 | 1:250 |
| Antibody | Anti-Mouse Alexa 488 | Jackson Inmunoresearch | 115-545-003 | 1:250 |
| Antibody | Anti-Rat Cy3 | Jackson Inmunoresearch | 112-165-167 | 1:250 |
| Antibody | Anti-Rat Alexa 697 | Jackson Inmunoresearch | 112-605-167 | 1:250 |
| Chemical compound | Sucrose | Sigma Aldrich | 102174662 | |
| Chemical compound | Caffeine | Sigma Aldrich | 102143502 | |
| Chemical compound | Agarose | Condalab | 8100.00 | |
| Chemical compound | Agarose (low melting) | Lonza | 50100 | |

## *Drosophila* husbandry

*Drosophila melanogaster* stocks were reared and maintained on standard 'Iberian' fly food under a 12 hr light:12 hr dark cycle at 25 °C. $w^{1118}$ and *Oregon-R* fly lines were used as mutant and wild-type strain controls unless otherwise indicated. Starvation was induced by transferring flies into vials containing a solid medium formed with a piece of paper (Kimberly-Clark Kimtech, 7552) soaked with 2.5 ml tap water. All *Drosophila* strains used in this thesis are listed in the key resource table.

## Tissue dissection and fluorescence activated cell sorting (FACS)

Central Brains or SEZs were dissected in cold Calcium- and Magnesium-free 1 X Dulbecco's Phosphate Buffered Saline (DPBS) (Thermo Fisher, 14190094), transferred to a Protein LoBind (Eppendorf, 0030108116) tube containing DPBS 0.01% BSA (Bovine Serum Albumin) (Invitrogen, AM2616) and digested with 1 mg/ml of papain and collagenase (Sigma-Aldrich, P4762 and C2674, respectively). After enzymatic digestion, the dissociated tissue solution was filtered through a 20 μm filter (pluriSelect, SKU 43-10020-40) into 300 μl DPBS BSA 0.01% in a Protein LoBind tube. Samples were kept at –80 °C or directly sorted by FACS.

Cells were sorted using a FACS AriaTM III. Several steps were followed to discard debris, clustered cells, and dead cells using a DAPI fluorescent filter. Gustatory Second Order Neurons labeled with mtdTomato were selected using a 555 nm laser (RFP PE-Texas Red A) to discard non-fluorescent cells

from the red fluorescent population. In addition to the 555 nm laser, to differentiate the autofluorescence (far red, $\lambda$ >670 nm) from the actual mtdTomato *trans*-Tango signal from gustatory second-order neurons ($\lambda$ =615 nm), a far red 566 nm laser (PE-Cy7-A) was applied consecutively to only conserve those mtdTomato fluorescent single cells for the PE-Texas Red-A laser and not for the PE-Cy7-A laser. All fluorescent gustatory second-order neurons were sorted into Protein LowBind RNase-free tubes containing 15 µl of lysis buffer for RNA extraction and finally stored at –80 °C. To sort the Lk GFP⁺ neurons, the 488 nm laser was used to sort only GFP⁺ fluorescent cells by applying the FITC-A filter. The total number of brains and cells dissected are indicated in *Supplementary file 1*.

## RNAseq analysis

### RNA extraction and sequencing

RNA was extracted using the PicoArcturus RNA isolation kit (Thermo Fisher, 12204–01). The spectrophotometer NanoDrop ND-1000 was used for samples estimated to contain higher amounts of total mRNA extracted, in the range of 100–3000 ng/µl. Additionally, to obtain the exact concentration and test the mRNA's quality, total mRNA was measured using the 2100 Bioanalyzer (Agilent Technologies). 50–5000 pg/µl range was measured using chips from the RNA 6000 Pico Kit (Agilent Technologies).

Samples were sequenced using 50 bp single reads on an *Illumina HiSeq2500* (for fed condition) and *Illumina NextSEq2000* (for starved condition) sequencers at the Genomics Unit of the Center for Genomic Regulation (Barcelona, Spain).

Reads were quality-checked with MultiQC v1.0 software using the tool FastQC v0.11.9. RNAseq output reads were aligned to the *D. melanogaster* reference genome from the Ensemble Project database (EMBL-EBI, Cambridge, UK) using STAR v2.7.9a (Spliced Transcripts Alignment to a Reference) (*Dobin et al., 2013*). Differential gene expression analysis was performed using DESeq2 v1.28.1 (*Love et al., 2014*) with the free *R* programming language (GNU project) software *RStudio* v4.2. Also, the reads within the gene were transformed by this tool to a total of counts per gene. Transcripts per million (TPM) were calculated using Salmon 1.10.2 (*Roberts et al., 2011*). A combination of packages (or libraries) from CRAN and Bioconductor v3.14 (*Gentleman et al., 2004*; *Huber et al., 2015*) open-source projects was used for data processing and plotting.

### Gene ontology analysis

The Gene Ontology (GO) analysis was performed using the open source interface PANGEA (Pathway, Network and Gene-set Enrichment Analysis; https://www.flyrnai.org/tools/pangea/) (*Hu et al., 2023*) specifically the *Drosophila* GO subsets terms at the online platform. Only those genes significantly up-regulated or significantly down-regulated in the differential gene expression analysis were included in the GO analysis. Each of the G2N's differentially expressed genes was analyzed separately, and then, only those GO terms of interest were plotted in a single graph.

Additionally, other GO analysis interfaces and tools as g::Profiler (https://biit.cs.ut.ee/gprofiler/gost), AmiGO2 (https://amigo.geneontology.org/amigo; *Ashburner et al., 2000*) undefined were used to validate the results of PANGEA. Finally, Gene set enrichment analysis (GSEA) was represented using RStudio v4.2 software (code available upon request).

## qPCR

Standard real-time quantitative PCR (qPCR) was performed with 2 ng of template cDNA, PowerUp SYBR Green PCR Master Mix (Applied Biosystems, A25742) and gene-specific primers read on QuantStudio 3 Real-Time PCR System (Applied Biosystems, A28567) with a standard cycling mode: 2 min at 50 °C and another 2 min at 95 °C followed by 40 x cycles of 15 s at 95 °C and 1 min at 60 °C. *Gapdh2* (Glyceraldehyde 3-phosphate dehydrogenase 2) of *D. melanogaster* protein expression levels was used as a housekeeping gene to normalize the results. Triplicate samples per each condition and technical triplicates were performed, and the relative gene expression was normalized by ΔCt analysis. Data is presented as mean ± SEM, and was statistically analyzed using two-tailed Student's *t*-test, considering $p < 0.05$ to be statistically significant. Primers used: *LK* (*Zandawala et al., 2018*) *Fw seq*: GCCTTTGGCCGTCAAGTCTA; *Rev seq*: TGAACCTGCGGTACTTGGAG; *Gapdh2* (*Uchizono et al., 2017*) *Fw seq*: CTACCTGTTCAAGTTCGATTCGAC; *Rev seq*: AGTGGACTCCACGATGTATTC.

## Immunohistochemistry

Immunofluorescence on peripheral and central tissues of adult flies was performed following standard procedures (*Croset et al., 2010*; *Sánchez-Alcañiz et al., 2017*). In brief, brains and proboscis were dissected in cold Phosphate Buffer (PB) and fixed for 25 min in 4% Paraformaldehyde (PFA EMS15710) in PB at RT. After that, brains and proboscis were washed five times for 10 min with PB +0.3% Triton X-100 (PBT) and blocked for 1 hr in PBT +5% normal goat serum (NGS) (Abcam, ab7481). Later, primary and secondary antibody (see Key resource table) incubations were for 48 hr each in PBT +5% NGS at 4 °C in constant agitation. Finally, after five washes of 10 min, brains and proboscis were equilibrated and mounted in Vectashield Antifade Mounting Medium with DAPI (Vector Laboratories, H-1200–10), maintaining their three-dimensional structure.

## Active-GRASP stimulation

### Sweet GRN stimulation

Pupae were selected and placed in tubes containing the desired stimuli to prevent adult flies from accessing our standard 'Iberian' food, which includes yeast and sucrose. *Gr64f-LexA, Lk-Gal4 >active-GRASP* pupae were transferred to tubes with a piece of paper (Kimberly-Clark Kimtech, 7552) soaked in tap water (control group, no activation of Gr64f$^{GRNs}$) or to a vial containing paper saturated with a 200 mM sucrose solution (sucrose stimulation group, activation of Gr64f$^{GRNs}$). Adults were maintained in each condition for 48 hr at 25 °C until dissection.

### Bitter GRN stimulation

*Gr66a-LexA; Lk-Gal4 >active-GRASP* 1–2 d-old adults were transferred into two different conditions: food with sucrose (control condition, no activation of Gr66a$^{GRNs}$) and food with sucrose and caffeine (bitter stimulation group, activation of Gr66a$^{GRNs}$). Adding sucrose to the food is necessary to induce the feeding of caffeine-containing food, as the bitter taste itself induces repulsion (*Wang et al., 2004*). Female flies were maintained for 3–4 d in each condition until dissection.

- Sucrose food: 200 mM sucrose (Sigma Aldrich, 102174662), miliQ water, 0,9% NaCl, 2,7% yeast, and 1% agarose.
- Sucrose +caffeine food: 200 mM sucrose, 75 mM caffeine, miliQ water, 0.9% NaCl, 2.7% yeast, and 1% agarose.

To enhance the stimulation of sweet and bitter GRNs, prior to dissection, flies were immobilized on P200 tips and the proboscis was exposed to a piece of paper (Kimberly-Clark Kimtech, 7552) containing 1 M Sucrose (for *Gr64f-LexA, Lk-Gal4 >active-GRASP* flies) and 75 mM Caffeine (for *Gr66a-LexA, Lk-Gal4 >active-GRASP* flies) for 1 hr in a humidified chamber. Following this, the flies were dissected according to the immunohistochemistry protocol.

## Microscopy image capture and processing

Images were acquired using a Leica SPEII laser scanning confocal microscope. Routinely, images of dimensions 512×512 pixels were acquired using an oil immersion ×20 objective in stacks of 1–2 μm. Files were saved in .*tiff* format and processed with ImageJ (Fiji) open-source image-processing software (*Schindelin et al., 2012*). For fluorescence quantification, the fluorescent intensity from the Region Of Interest (ROI) area was analyzed by using the ROI Manager from ImageJ, and data is presented as mean ± standard error of the mean (SEM) and was statistically analyzed using two-tailed Student's *t*-test, considering $p < 0.05$ to be statistically significant.

## Identification of the SELK neurons in the adult brain of *Drosophila* connectome

Neurons were reconstructed in a serial section transmission electron volume (*Zheng et al., 2018*) using FlyWire (https://flywire.ai/) (*Dorkenwald et al., 2022*; *Dorkenwald et al., 2024*; *Schlegel et al., 2023*). To accomplish this, 52 sweet and bitter GRNs skeletons (.*swc* format), previously described by *Engert et al., 2022*, were downloaded from the open-source interface of Virtual Fly Brain (VFB) (*Milyaev et al., 2012*) and aligned to the Full Adult Fly Brain (FAFB) using the JRC2018U reference brain as a template (*Zheng et al., 2018*; *Bogovic et al., 2020*; *Costa et al., 2016*). Among these, 19 GRNs were bitter sensing (12 from group 1 (*Figure 6—figure supplement 1A*) and seven from

group 2 (*Figure 6—figure supplement 1B*)), while 33 GRNs were sweet sensing (17 from group 4 (*Figure 6—figure supplement 1C*) and 16 from group 5 (*Figure 6—figure supplement 1D*)). Only Gr64f^GRNs and Gr66a^GRNs from the right hemisphere were used in this analysis due to the larger dataset. All these GRN skeletons were aligned to the recently released FlyWire dataset, which is a dense, machine learning-based reconstruction of over 80,000 FAFB neurons (*Dorkenwald et al., 2024*). Using the Flywire Gateway tool, each of the sweet and bitter GRNs analyzed was aligned and reconstructed to the EM dataset in the FlyWire toolbox, with the IDs annotated (*Dorkenwald et al., 2022*).

We next used BrainCircuits (*Gerhard et al., 2023*) to identify all postsynaptic partners to sweet Gr64f^GRNs and bitter Gr66a^GRNs. The postsynaptic candidates were downloaded without limiting the number of synaptic contacts among neurons. We noted 12,594 postsynaptic hits for Gr66a^GRNs and 36790 hits for Gr64f^GRNs. By intersecting both datasets, we obtained 333 downstream segment IDs as potential G2Ns receiving inputs from both GRNs. We refined our list to 17 candidates through visual inspection, focusing on neurons with projections within the SEZ (*Supplementary file 2*).

In order to identify which of the 17 candidates presented in *Supplementary file 2* correspond to the SELK neurons, we proceeded to register the SELK neurons in a reference brain template. To achieve this, immunohistochemistry was performed on 7-d-old *Lk-Gal4 >UAS-CD8::GFP* flies using nc82 (to label the brain structure) and GFP (to label the SELK neurons). Images were taken at a resolution of 1024×1024 and aligned to the 'JRC2018U' reference brain template (*Bogovic et al., 2020*) using CMTK software in Fiji (for more details, see Flywire self-guided training). The aligned images were used to create a skeleton of the SELK neurons using the Single Neurite Tracer (SNT) plugin in Fiji (for more details, see Flywire self-guided training), which was then saved in .*swc* format. Only the most labeled trace of the SELK neuron (including the soma and main axon) was included in the skeleton to avoid false positive tracings. Finally, the SELK neuron skeletons were aligned to the FlyWire dataset (*Dorkenwald et al., 2022*) using the Flywire Gateway tool, taking the JRC2018U reference brain (*Bogovic et al., 2020*) as a template. The resulting point cloud in the FAFB of the FlyWire Sandbox was used to guide a search for nearby neurons to match the possible SELK neuron candidates' IDs previously identified.

## Behavior

### Proboscis extension reflex (PER)

PER to labellar stimulation was assessed following a standard protocol (*Shiraiwa and Carlson, 2007*). Flies were anesthetized on ice and individually immobilized on P200 tips, whose narrow end was cut so that only the fly's head could protrude from the opening, leaving the rest of the body, including legs, constrained inside the tip. After flies were recovered in a humidified chamber for 20 min, flies were water-satiated before testing ad libitum, discarding those that, after 5 min, continued extending their proboscis in response to water. Tastants were delivered using a small piece of paper (Kimberly-Clark Kimtech, 7552) and touching very gently the labellum for 2 s maximum, leaving a gap of 1 min between stimulations. Sucrose (Sigma Aldrich, 102174662) and caffeine (Sigma-Aldrich, 102143502) were diluted in miliQ sterile water appropriately. PER was manually recorded: only full proboscis extensions were counted as PER and registered as 1, considering partial or absent proboscis extensions as 0. Finally, flies were tested at the end of the experiment with water as the negative control and 1 M of sucrose as the positive control. Only flies that showed negative PER for water and positive PER for 1 M of sucrose were included in the analysis.

### flyPAD two-choice assay

flyPAD assays were performed to study the feeding microstructure in a two-choice feeding paradigm as described previously (*Itskov et al., 2014*), and several feeding parameters were measured individually in a high-throughput manner. Individual flies were placed in individual arenas with two different food sources in independent well electrodes. The flyPAD hardware used comprised 56 individual chambers divided into two independent pieces of hardware, each consisting of 28 chambers connected to an independent computer. All tastants used were solved in water 1% low-melting agarose (Lonza, 50100). To transfer files to each chamber, flies were anesthetized in ice for 5 min and individually placed in the arena by mouth aspiration according to its sex and genotype. All the experiments began once all the flies woke up and were active. Flies were allowed to feed at 25 °C for 60 min. After time ended, dead flies were annotated. flyPAD data were acquired using the Bonsai

framework, and analyzed in MATLAB using custom-written software delivered by *Itskov et al., 2014*. Then, specific *R* software scripts developed by the lab were applied to analyze the bulk of the flyPAD experiment data (https://gitlab.com/sanchez-alcaniz_lab/molla-albaladejo copy archived at *Sanchez-Alcaniz, 2025*). Only those flies that performed more than 2 bouts and 25 sips were included in the analysis.

## Statistical analysis

The sample size was determined based on preliminary experiments. Data were analyzed using *R* software v4.1.0 (R Foundation for Statistical Computing, Vienna, Austria, 2005; https://www.r-project.org) and plotted using *R* or *GraphPad Prism 6*. The statistical test is determined in the figure legend for each of the plots. The Bonferroni method was used when p-value correction for multiple comparisons was required. Except for PER and flyPAD experiments, quantitative data show their distribution by superimposing a boxplot. For the boxplots, the whiskers are calculated as follows: the upper whisker equals the third quartile plus 1.5x the interquartile range (IQR), and the lower whisker equals the first quartile minus 1.5x the IQR. Any data points above the superior or below the inferior whisker values are considered outliers. The outliers were included in the statistical comparisons as we performed non-parametric rank tests.

For PER experiments, data were analyzed using a generalized linear model based on a logistic regression set to a binomial distribution model (function glm () in *R* software), where the response of the flies is dependent on their genotype responding to different stimuli of sucrose and caffeine (glm() equation used below). Error bars represent the standard error of the proportion ($\sqrt{p\left(1-p\right)/n}$).

$$glm(Response \sim Genotype^{*}Stimulus, data = PERdata, family = binomial)$$

For flyPAD experiments, a set of *R* software scripts developed by the lab was used to analyze the feeding microstructure behavior, which employs different statistical tests depending on the final goal of the analysis. The preference index is calculated as follows:

$$Preference\ Index\ (PI) = \frac{N_A - N_B}{N_{TOTAL}}$$

## Acknowledgements

We thank Prof. Richard Benton, Prof. Roman Arguello, and members of the JSA laboratory for their comments on the manuscript. We also thank our colleagues from the Institute of Neuroscience (IN) for their insightful inputs while developing the present research. This work was funded by the Spanish State Research Agency (AEI/10.13039/501100011033), operating grants PID2019-105839GA-I00, CNS2022-135109, 'Ramón y Cajal' Fellowship RyC2019-026747-I and 'Severo Ochoa' Centre of Excellence grant to the IN (grant CEX2021-001165-S).

## Additional information

### Funding

| Funder | Grant reference number | Author |
|---|---|---|
| Generalitat Valenciana | CIDEGENT/2018/035 | Juan Antonio Sanchez-Alcaniz |
| Spanish National Plan for Scientific and Technical Research and Innovation | PRE-2020-095035 | Manuel Jiménez-Caballero |
| Spanish National Plan for Scientific and Technical Research and Innovation | PID2019-105839GA-I00 | Juan Antonio Sanchez-Alcaniz |

| Funder | Grant reference number | Author |
|---|---|---|
| Spanish National Plan for Scientific and Technical Research and Innovation | RyC2019- 026747-I | Juan Antonio Sanchez-Alcaniz |
| Spanish State Research Agency | CNS2022-135109 | Juan Antonio Sanchez-Alcaniz |

The funders had no role in study design, data collection and interpretation, or the decision to submit the work for publication.

## Author contributions
Rubén Mollá-Albaladejo, Conceptualization, Data curation, Software, Formal analysis, Visualization, Methodology, Writing – original draft; Manuel Jiménez-Caballero, Software, Methodology; Juan Antonio Sanchez-Alcaniz, Conceptualization, Resources, Data curation, Software, Formal analysis, Supervision, Funding acquisition, Validation, Investigation, Visualization, Methodology, Writing – original draft, Project administration, Writing – review and editing

## Author ORCIDs
Rubén Mollá-Albaladejo  http://orcid.org/0000-0001-9897-9525
Juan Antonio Sanchez-Alcaniz  https://orcid.org/0000-0002-4900-5086

Reviewer #1 (Public review): https://doi.org/10.7554/eLife.100947.3.sa1
Reviewer #2 (Public review): https://doi.org/10.7554/eLife.100947.3.sa2
Reviewer #3 (Public review): https://doi.org/10.7554/eLife.100947.3.sa3
Author response https://doi.org/10.7554/eLife.100947.3.sa4

# Additional files

## Supplementary files
Supplementary file 1. Number of brains dissected and cells sorted for the RNAseq experiment.

Supplementary file 2. List of candidate IDs.

MDAR checklist

## Data availability
The transcriptomic data generated in this study have been deposited in the ArrayExpress database under accession code E-MTAB-15220. Code for flyPAD analysis can be downloaded using this link: https://gitlab.com/sanchez-alcaniz_lab/molla-albaladejo copy archived at *Sanchez-Alcaniz, 2025*.

The following dataset was generated:

| Author(s) | Year | Dataset title | Dataset URL | Database and Identifier |
|---|---|---|---|---|
| Sanchez-Alcaniz J | 2025 | RNAseq of *Drosophila* Sweet and Bitter Gustatory Second Order Neurons | https://www.ebi.ac.uk/biostudies/arrayexpress/studies/E-MTAB-15220 | E-MTAB-15220, ArrayExpress |

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
