## [Editor Report · eLife Assessment]

This study provides **valuable** insights into the organization of second-order circuits of gustatory neurons, particularly in how these circuits integrate opposing taste inputs and are modulated by metabolic state to regulate feeding behavior. Through an elegant combination of complementary techniques, the authors identify the target neurons involved in gustatory integration. The evidence supporting their conclusions is **convincing**.

---

## [Referee Report · Reviewer #1 (Public review)]

Summary:

Mollá-Albaladejo et al. investigate the neurons downstream of GR64f and Gr66a, called G2Ns. They identify downstream neurons using trans-Tango labeling with RFP and then perform bulk RNA-seq on the RFP-sorted cells. Gene expression is up- or downregulated between the cell populations and between fed and starved states. They specifically identify Leukocinin as a neuropeptide that is upregulated in starved Gr66a cells. Leucokinin cells, identified by a GAL4 line, indeed show higher expression when starved, especially in the SEZ. Furthermore, Leucokinin cells colocalize with the trans-Tango signal from downstream neurons of both GRs. This connection is confirmed with GRASP and active GRASP. According to EM data, Leucokinin cells in the SEZ receive a lot of input and connect to many downstream neurons. In behavior experiments performed with flies lacking Leucokinin neurons, flies show reduced responsiveness to sugar and bitter mixtures when starved. The authors suggest that Leucokinin neurons integrate bitter and sugar tastes and that their output is modified by a hunger state.

Strengths:

The authors use a multitude of tools to identify SELK neurons downstream of taste sensory neurons and as starvation-sensitive cells. This study provides an example of how combining genetic labeling, RNA-seq, and EM analysis can be used to investigate the function of specific neural circuits.

Weaknesses:

The authors now provide more evidence to show a functional connection between sensory neurons and SELK neurons, for example, by using active GRASP, however, different staining methods reveal different connectivity patterns.The authors describe a behavioral phenotype when flies are starved, however, the phenotype can still not clearly be assigned to the SELK neurons.

---

## [Referee Report · Reviewer #2 (Public review)]

Summary:

A core task of the brain is processing sensory cues from the environment. The neural mechanisms of how sensory information is transmitted from peripheral sense organs to subsequent being processing in defined brain centers remains an important topic in neuroscience. The taste system hereby assesses the palatability of food by evaluating the chemical composition and nutrient content while integrating the current need of energy by assessing the satiation level of the organism. The current manuscript provides insights into the early circuits gustatory coding using the fruit fly as model. By combining trans-tango and FACS-based bulk RNAseq to assess the target neurons of sweet sensing (using by Gr64f-Gal4) and bitter sensing (using Gr66a-Gal4) in a first set of experiments the authors investigate genes that are differentially expressed or co-expressed in normal and starved conditions. With a focus on neuropeptides and neurotransmitters differential expression in the different conditions were assessed resulting in the identification of Leucokinin as potentially interesting gene. The notion is further supported by RNAseq of Lk-Gal4>mCD8:GFP sorted cells and immunostainings. GRASP and BacTrace experiments further supports that the two Lk expressing cells in the SEZ should indeed be postsynaptic to both type of sensors. Using EM-based connectomics data (based on a previous publication by Engert et al.), the authors also look for downstream targets of the bitter versus sweet gustatory neurons to identify the Lk-neurons. Based on morphology they identify candidates and further depict the potential downstream neurons in the connectome, which appears largely in agreement with GRASP experiments. Finally silencing the Lk-neurons shows an increased PER response in starved flies (when combined with bitter compounds) as well as increased feeding in a FlyPad assay.

Strengths:

Overall this is an intriguing manuscript, which provides insight into the organization of 2nd order gustatory neurons. It specifically provides strong evidence for the Lk-neurons as target of sweet and bitter GRNs and provides evidence for their role in regulating sweet vs bitter based behavioral responses. Particularly the integration of different techniques and datasets in an elegant fashion is a strong side of the manuscript. Moreover to put the known LK-neurons into the context of 2nd order gustatory signalling is strengthening the knowledge about this pathway.

Weaknesses:

I do not see any major weakness in the current manuscript. Novelty is to some degree lessened by the fact, that the RNAseq approach did not identify new neurons but rather put the known LK-neurons as major finding. Similarly the final behavioral section is not very deep and to some degree corroborates the previous publication by the Keene and Nässel labs- that said, the model they propose is indeed novel (but lacks depth in analyses, e.g. there is no physiology that would support the modulation of Lk neurons by either type of GRN). The connectomic section appears a bit out of place and after reading it it's not really clear what one should make of the potential downstream neurons (particularly since the Lk-receptor expression has been previously analyzed); here it might have been interesting to address if/how Lk-neurons may signal directly via a classical neurotransmitter (an information that might be found easily in the adult brain single-cell data).

Comments on the latest version:

I feel all points have been included to a satisfactory degree.

---

## [Referee Report · Reviewer #3 (Public review)]

Summary:

To make feeding decisions, animals need to process three types of information: positive cues like sweetness, negative cues like bitterness, and internal states such as hunger or satiety. This study aims to identify where the information is integrated in the fruit fly brain. The authors applied RNA sequencing on second-order gustatory neurons responsible for sweet and bitter processing, under fed and starved conditions. The sequencing data reveal significant changes in gene expression across sweet vs. bitter pathways and fed vs. starved states. The authors focus on the neuropeptide Leucokinin (Lk), whose expression is dependent on the starvation state. They identify a pair of neurons, named SELK neurons, which express Lk and receive direct input from both sweet and bitter gustatory neurons. These SELK neurons are ideal candidates to integrate gustatory and internal state information. Behavioral experiments show that blocking these neurons in starved flies alters their tolerance to bitter substances during feeding.

Strengths:

(1) The study employs a well-designed approach, targeting specific neuronal populations, which is more efficient and precise compared to traditional large-scale genetic screening methods.

(2) The RNAseq results provide valuable data that can be utilized in future studies to explore other molecules beyond Lk.

(3) The identification of SELK neurons offers a promising avenue for future research into how these neurons integrate conflicting gustatory signals and internal state information.

Weaknesses:

Unfortunately, due to technical challenges, the authors were unable to directly image the functional activity of SELK neurons.

---

## [Author Response]

The following is the authors’ response to the original reviews.

**Public Reviews:**

**Reviewer #1 (Public review):**
Summary:Mollá-Albaladejo et al. investigate the neurons downstream of GR64f and Gr66a, called G2Ns. They identify downstream neurons using trans-Tango labeling with RFP and then perform bulk RNA-seq on the RFP-sorted cells. Gene expression is up- or downregulated between the cell populations and between fed and starved states. They specifically identify Leukocinin as a neuropeptide that is upregulated in starved Gr66a cells. Leucokinin cells, identified by a GAL4 line indeed show higher expression when starved, especially in the SEZ. Furthermore, Leucokinin cells colocalize with the transTango signal from downstream neurons of both GRs. This connection is confirmed with GRASP. According to EM data, Leucokinin cells in the SEZ receive a lot of input and connect to many downstream neurons. In behavior experiments performed with flies lacking Leucokinin neurons, flies show reduced responsiveness to sugar and bitter mixtures when starved. The authors suggest that Leucokinin neurons integrate bitter and sugar tastes and that their output is modified by a hunger state.Strengths:The authors use a multitude of tools to identify SELK neurons downstream of taste sensory neurons and as starvation-sensitive cells. This study provides an example of how combining genetic labeling, RNA-seq, and EM analysis can be combined to investigate neural circuits.Weaknesses:The authors do not show a functional connection between sensory neurons and SELK neurons. Additionally, data from RNA seq, anatomical studies, and EM analysis are sometimes contradictory in terms of connectivity. GRASP signal is not foolproof that cells are synaptically connected.

We appreciate the reviewer’s comments. Unfortunately, we have not successfully demonstrated a functional response of SELK neurons using in vivo calcium imaging with UAS-GCaMP7 (we tried f, m, and s versions), primarily due to challenges in obtaining stable signals. We stimulated GRNs using sucrose, caffeine, or a mixture of both, and maybe even if the concentrations were high, they were not enough to induce a response.

Regarding GRASP, we acknowledge its limitations as a standalone technique for establishing genuine synaptic connections between neurons, as some signals may reflect false positives resulting from the mere proximity of the candidate neurons. To strengthen our findings, we complemented these results by demonstrating the positive colocalization of the Leucokinin antibody signal over the *Gr66aGal4>trans-TANGO* and *Gr64f-Gal4>trans-TANGO* (Figure 4), confirming that Leucokinin neurons are indeed postsynaptic to both sweet and bitter GRNs. Moreover, we incorporated BacTrace data to highlight the direct connectivity between sweet and bitter GRNs (now Figure 5E).

In the revised manuscript, we have introduced the active-GRASP technique (Macpherson et al., 2015). In this version of GRASP, the presynaptic half of GFP (GFP 1-10) is fused to synaptobrevin, which becomes accessible in the membrane of the presynaptic neuron within the synaptic cleft upon presynaptic stimulation (in our case, by stimulating with sucrose sweet Gr64f^GRNs^ and with caffeine the bitter Gr66a^GRNs^). Utilizing this technique, we successfully demonstrated (see new Figure 5B and 5D) that when presented with water, no signal was detected in the *Gr66a-LexA, Lk-Gal4 > active-GRASP*, or *Gr64f-LexA, Lk-Gal4 > active-GRASP* transgene flies. However, in the presence of caffeine, *Gr66aLexA, Lk-Gal4 > active-GRASP* transgene flies exhibited a clear signal in the SEZ, and similarly, sucrose presentation to *Gr64f-LexA, Lk-Gal4 > active-GRASP* transgene flies yielded a detectable signal. The results obtained from *active-GRASP* provide additional evidence supporting the connectivity between SELK neurons and both Gr64f^GRNs^ and Gr66a^GRNs^, further indicating the functional connectivity of the GRNs and SELK neurons.

The authors describe a behavioral phenotype when flies are starved, however, they do not use a specific driver for the described cell type, thus they should also tone down their claims.

We agree with the reviewer that the Lk-Gal4 driver line used labels SELK, LHLK, and ABLK neurons. The behavior examined in this paper, the Proboscis Extension Response (PER), measures the initiation of feeding. Although the neural circuit involved in this behavior is primarily confined to the SEZ where SELK neurons are located, we cannot rule out the possibility that other Lk neurons may also play a role in the process. To restrict expression of the Tetanus Toxin, we have utilized the *tsh-Gal80* (Clyne et al., 2008) transgene in combination with the *Lk-Gal4>UAS-TNT* and *Lk-Gal4>UAS-TNTimp* constructs to prevent the expression of the Tetanus Toxin in ABLK neurons, thereby restricting its expression to the SELK and LHLK neurons in the central brain. The new results (Sup Figure 7A) indicate that ABLK neurons do not play a role in integrating sweet and bitter information. However, we acknowledge the reviewer's point that we are still silencing LHLK neurons, so we have adjusted our claims to align more closely with our data

Generally, the authors do not provide a big advancement to the field and some of the results are contradictory with previous publications.

We believe our work does not contradict previous findings, nor does it invalidate the role of ABLK neurons in water homeostasis or the role of LHLK neurons in regulating sleep via starvation. We provide additional information on the possible role of SELK neurons in integrating gustatory information. The location of SELK neurons in the SEZ suggests that they may play a role in feeding behavior, and we have demonstrated that these neurons are indeed involved in integrating gustatory information to influence feeding decisions. We consider we have contributed by highlighting a new role for the Leucokinin neuropeptide in feeding behavior.

**Reviewer #2 (Public review):**
Summary:A core task of the brain is processing sensory cues from the environment. The neural mechanisms of how sensory information is transmitted from peripheral sense organs to subsequent being processing in defined brain centers remain an important topic in neuroscience. The taste system hereby assesses the palatability of food by evaluating the chemical composition and nutrient content while integrating the current need for energy by assessing the satiation level of the organism. The current manuscript provides insights into the early circuits of gustatory coding using the fruit fly as a model. By combining trans-tango and FACS- based bulk RNAseq to assess the target neurons of sweet sensing (using Gr64fGal4) and bitter sensing (using Gr66a-Gal4) in a first set of experiments the authors investigate genes that are differentially expressed or co-expressed in normal and starved conditions. With a focus on neuropeptides and neurotransmitters, different expressions in the different conditions were assessed resulting in the identification of Leucokinin as a potentially interesting gene. The notion is further supported by RNAseq of Lk- Gal4>mCD8:GFP sorted cells and immunostainings. GRASP and BacTrace experiments further support that the two Lk- expressing cells in the SEZ should indeed be postsynaptic to both types of sensories. Using EM-based connectomics data (based on a previous publication by Engert et al.), the authors also look for downstream targets of the bitter versus sweet gustatory neurons to identify the Lk-neurons. Based on the morphology they identify candidates and further depict the potential downstream neurons in the connectome, which appears largely in agreement with GRASP experiments. Finally silencing the Lk- neurons shows an increased PER response in starved flies (when combined with bitter compounds) as well as increased feeding neurons shows an increased PER response in starved flies (when combined with bitter compounds) as well as increased feeding in a FlyPad assay. Strengths:Overall this is an intriguing manuscript, which provides insight into the organization of 2nd order gustatory neurons. It specifically provides strong evidence for the Lk-neurons as a target of sweet and bitter GRNs and provides evidence for their role in regulating sweet vs bitter-based behavioral responses. Particularly the integration of different techniques and datasets in an elegant fashion is a strong side of the manuscript. Moreover to put the known LK-neurons into the context of 2nd order gustatory signalling is strengthening the knowledge about this pathway.Weaknesses:I do not see any major weakness in the current manuscript. Novelty is to some degree lessened by the fact, that the RNAseq approach did not identify new neurons but rather put the known LK-neurons as major findings. Similarly, the final behavioral section is not very deep and to some degree corroborates the previous publication by the Keene and Nässel labs - that said, the model they propose is indeed novel (but lacks depth in analyses; e.g. there is no physiology that would support the modulation of Lk neurons by either type of GRN). The connectomic section appears a bit out of place and after reading it it's not really clear what one should make of the potential downstream neurons (particularly since the Lk-receptor expression has been previously analyzed); here it might have been interesting to address if/how Lk-neurons may signal directly via a classical neurotransmitter (an information that might be found easily in the adult brain single-cell data).

We thank the reviewer for the comment. Indeed, we attempted in vivo Ca imaging but were unsuccessful. We have rewritten the connectomic section to better integrate it with the rest of the text and have reanalyzed the data obtained. We considered gathering data from the single-cell adult dataset, but this dataset includes the entire adult fly brain, encompassing SELK and LHLK neurons, making it impossible to differentiate between the two types of Lk neurons. Any further analysis will require transcriptomic analysis of SELK via scRNAseq under the different metabolic conditions tested in this study work.

**Reviewer #3 (Public review):**
Summary:To make feeding decisions, animals need to process three types of information: positive cues like sweetness, negative cues like bitterness, and internal states such as hunger or satiety. This study aims to identify where the information is integrated into the fruit fly brain. The authors applied RNA sequencing on second-order gustatory neurons responsible for sweet and bitter processing, under fed and starved conditions. The sequencing data reveal significant changes in gene expression across sweet vs. bitter pathways and fed vs. starved states. The authors focus on the neuropeptide Leucokinin (Lk), whose expression is dependent on the starvation state. They identify a pair of neurons, named SELK neurons, which express Lk and receive direct input from both sweet and bitter gustatory neurons. These SELK neurons are ideal candidates to integrate gustatory and internal state information. Behavioral experiments show that blocking these neurons in starved flies alters their tolerance to bitter substances during feeding.Strengths:(1) The study employs a well-designed approach, targeting specific neuronal populations, which is more efficient and precise compared to traditional large-scale genetic screening methods.(2) The RNAseq results provide valuable data that can be utilized in future studies to explore other molecules beyond Lk.(3) The identification of SELK neurons offers a promising avenue for future research into how these neurons integrate conflicting gustatory signals and internal state information.Weaknesses:(1) Unfortunately, due to technical challenges, the authors were unable to directly image the functional activity of SELK neurons.(2) In the behavioral experiments, tetanus toxin was used to block SELK neurons. Since these neurons may release multiple neurotransmitters or neuropeptides, the results do not specifically demonstrate that Leucokinin (Lk) is the critical factor, as suggested in Figure 8. To address this, I recommend using RNAi to inhibit Lk expression in SELK neurons and comparing the outcomes to wild-type controls via the PER assay.

We appreciate the author's comments and suggestions. As noted, Tetanus Toxin silences the neuron’s activity, affecting the functioning of various neurotransmitters and neuropeptides released by the targeted neuron. In response to the reviewer's recommendation, we employed an RNAi line specifically designed to silence Leucokinin production in Lk-expressing neurons.

The results presented in Supplementary Figure 7B demonstrate that knocking down Leucokinin in Lk neurons significantly reduces the flies' tolerance to caffeine in sweet food.

It is crucial to highlight that the sucrose concentration used in Figure 7C was 50mM, whereas in Supplementary Figure 7B, it was increased to 100mM. This adjustment was necessary because the *Lk-Gal4*, *UAS-RNAi*, and *Lk-Gal4>UAS-RNAi* transgenic lines exhibited reduced sensitivity to sucrose compared to the *Lk-Gal4>UAS-TNT* or *Lk-Gal4>UAS-TNTimp* lines. We aimed to establish a sucrose concentration that would elicit a 50% Proboscis Extension Response (PER) without adding any other compound, thereby allowing us to evaluate the additional effect of caffeine in the food.

However, according to the data derived from the connectome, SELK neurons might be cholinergic, and this neurotransmitter might be involved in controlling also the behavior of the flies.

**Recommendations for the authors:**

**Reviewer #1 (Recommendations for the authors):**
To get more evidence for connections between sensory cells and SELK neurons, could the authors also analyze a second available EM data set? Would setting a different threshold (>5 synapses) reveal connections to both sensories? Comparisons between SELK in- and outputs from EM data and Tango labeling also seem to differ quite a lot based on provided images - can the authors count cell bodies in the stainings? Further proof would be to provide functional imaging data that shows that SELK neurons respond to sugar and bitter compounds.

In this study, we utilized the recently published EM dataset for the *Drosophila* central brain connectome (Dorkenwald et al., 2024; Flywire.ai). Changing the number of synapses affects the counts of pre- and postsynaptic neurons. We set a threshold of more than five synapses, as recommended by Flywire, to avoid false positives (Dorkenwald et al., 2024). This threshold has been widely used in recent papers (Engert et al., 2022; Shiu et al., 2022; Walker et al., 2025).

The neuron counts in the connectomic data differ from those in the *trans-* and *retro-TANGO* experiments. In our initial *trans-TANGO* experiment, which labeled postsynaptic neurons in the Gr64fGal4 and Gr66a-Gal4 transgenic lines, we counted the labeled neurons (see Supplementary Figure 1C) and observed considerable variability between different brains. Due to anticipated variability, we did not count the labeled neurons from *trans-TANGO* and *retro-TANGO* techniques in the Leucokinin neurons. Furthermore, neither technique labels all postsynaptic or presynaptic neurons, respectively. A recent study on the *retro-TANGO* technique (Sorkac et al., 2023) found a minimum threshold: the presynaptic neuron must form a certain number of synapses with the neuron of interest to be adequately labeled. According to this paper, the established threshold is 17 synapses. It is likely that the *trans-TANGO* technique also has a threshold relating to the number of labeled neurons, contingent on the synapse count. This would explain the discrepancy between the two results.

Unfortunately, we have not been able to provide functional data pointing to the activation of SELK neurons by sucrose or caffeine. However, our active-GRASP data indicates that the connectivity between Gr64f^GRNs^ and Gr66a^GRNs^ with SELK neurons is present and functional.

How many Leucokinin-positive cells are in the SEZ? Does the RNA-seq data provide further information about the SELK neurons? Potential receptor candidates for how they integrate hunger signals? AMPKa was described to be required in LHLK neurons.

There are two SELK neurons in the SEZ. Due to the nature of our bulk RNA sequencing (RNAseq), we cannot link any additional gene expressions detected in our transcriptomic analysis specifically to the SELK neurons regarding the integration of various signaling processes. Furthermore, the single-cell RNA sequencing (scRNAseq) data available from the *Drosophila* brain, as reported by Li et al. (2022), does not allow accurate differentiation between SELK and LHLK neurons. To understand how these neurons integrate both metabolic and sensory information, it is crucial to conduct a focused RNAseq study specifically on the SELK neurons to understand how these neurons integrate both metabolic and sensory information. This targeted analysis would provide the necessary insights to elucidate their functional roles better. However, according to the data derived from the connectome, SELK neurons might be cholinergic, and this neurotransmitter might be involved in controlling also the behavior of the flies.

According to previous studies (Yurgel et al., 2019), the Lk-GAL4 line is also expressed in the VNC, thus the authors could make use of the tsh-GAL80 tool to clean up the line. This study also performed GCaMP imaging in fed and 24h starved animals in SELK and couldn't find a difference, can the authors explain this discrepancy?

We thank the reviewer for this suggestion. We have now added a new piece of data using the *tsh-Gal80* transgene in our PER experiments (Supplementary Figure 7A). Blocking the expression of TNT in the ABLK neurons does not affect the main conclusion of the behavioral results. As stated previously, we were unable to obtain in vivo Ca imaging responses in SELK neurons upon exposure to sucrose, caffeine, or mixtures of sucrose and caffeine. We do not believe this is a discrepancy with previous works like Yurgel et al., 2019. It is likely that we faced technical issues regarding expression stability and that the stimulation was possibly too weak to detect changes in GFP levels

**Reviewer #2 (Recommendations for the authors):**
As mentioned above I do not have any major comments on the manuscript, but there are a few points that I feel should be considered:(1) The identification of the Lk-candidate neurons in the connectome remains a bit mysterious. In the method sections, this reads as follows "manual and visual criteria were applied to identify the neurons of interest ". (a) What precisely was done to get to the candidates? (b) Are there alternative candidates that may be Lk-neurons? (c) How would another neuron affect the conclusion of the downstream analysis?

We thank the reviewer for this comment. We have now modified and added new information in the connectomic section, reinforcing our conclusions and correcting the results obtained.

Our GRASP, BacTRace, and immunohistochemistry experiments pointed to SELK neurons as postsynaptic to both Gr64f^GRNs^ (sweet) and Gr66a^GRNs^ (bitter). To identify which neurons in the connectome could be the SELK neurons, we utilized a previously described set of GRNs already identified in the connectome (Shiu et al., 2022). We extracted all postsynaptic neurons to the sweet and bitter GRNs identified and intersected both datasets, retaining only those candidate hits receiving simultaneous input from sweet and bitter GRNs. This process yielded a total of 333 hits. Through visual inspection, we discarded all hits that were merely neuronal fragments or neurons that clearly were not our candidates. We narrowed the list down to a final set of 17 candidate neurons whose arborization was located in the SEZ. We reduced the candidates to two final entries from this list: ID 720575940623529610 (GNG.276) and ID 720575940630808827 (GNG.685). The GNG.276 neuron had a counterpart in the SEZ identified as GNG.246. Both of these neurons were annotated as DNg70 in the Flywire database. GNG.685 had a counterpart identified as GNG.595, and these two neurons were classified as DNg68. In both cases, the neuronal candidates, DNg70 and DNg68, were classified as descending neurons, a characteristic of previously described SELK neurons (Nässel et al., 2021). In our initial analysis published in bioRxiv and sent for revision, we identified DNg70 as potentially the SELK neurons based solely on the morphology of the neurons via visual inspection. However, we employed a better method to determine which candidate is more likely to be the SELK neurons, concluding that DNg68, rather than DNg70, represents the SELK neurons. Briefly, we performed an immunohistochemistry for GFP in the *Lk-Gal4>UAS-CD8:GFP* flies. We aligned the resulting image in a *Drosophila* reference brain (JRC2018 U) using the CMTK Registration plugin in ImageJ. The resulting image was skeletonized using the Single Neurite Tracer plugin in ImageJ and later uploaded to the Flywire Gateway platform to compare the structure of the aligned and skeletonized SELK neurons to our candidates. This comparison clearly indicated that the DNg68 neurons are the best candidates for representing the SELK neurons, rather than DNg70. We have updated the text and Figures 6 and Supplementary Figure 6 to reflect the new results. These new results do not alter the conclusions of the paper.

(2) In the transcriptomic experiments It seems that the raw transcripts are reporters, rather than normalised data. Why?

All transcriptomic data is normalized. In Figure 1 the differential expression was calculated using Deseq2 normalized counts. In Figure 2, Transcripts Per Million (TPM) were calculated using the Salmon package and normalized for the gene length.

(3) The expression of nAChRbeta1 in the transcriptomic data is rather striking. However, this remains currently not addressed: is this expression real?

We have not confirmed the upregulation or downregulation in gene expression for other but for Leucokinin, which is our main interest. We found the presence of nAChRbeta1 interesting, as GRNs are cholinergic (Jaeger et al., 2018), suggesting that it would make sense to find cholinergic receptors in G2Ns. However, it is possible that these receptors are expressed in all G2Ns and serve as a common means of communication.

(4) The description of the behavioural experiments in the results section is rather brief. I had a hard time following it since the genotypes are not repeated nor is it stated what is different in the experimental group vs control (but instead simply what changes in the experimental group, in a rather discussion-like fashion).

We thank the reviewer for the comment, we have rewritten this section to improve its clarity.

(5) If I understand the genetics for the behavioural experiments correctly it addresses the entire Lk-Gal4 expressing population, thus it is not possible to describe the role of the two SEZ neurons, but rather LkGal4 neurons. This should be clarified.

We thank the reviewer for this comment. Indeed, the Lk-Gal4 driver we used drives expression in all Leucokinin neurons, making it impossible to distinguish between the SELK, LHLK, or ABLK neurons. We have added a new piece of behavioral data by using the tsh-Gal80 transgene to prevent the expression of TNT in the ABLK neurons (Supplementary Figure 7A), but still we cannot distinguish between SELK and LHLK. We have rewritten the text to clarify this fact.

**Reviewer #3 (Recommendations for the authors):**
Overall, the manuscript is well-written, I only have one minor suggestion for improvement. In Figure 8C, please clarify the use of TNT to block Lk release.

We thank the reviewer for the comment, we have clarified the use of TNT in the text.

ReferencesClyne, J. D. & Miesenböck, G. Sex-Specific Control and Tuning of the Pattern Generator for Courtship Song in *Drosophila*. Cell 133, 354–363 (2008).

Dorkenwald, S. et al. Neuronal wiring diagram of an adult brain. Nature 634, 124–138 (2024).

Engert, S., Sterne, G. R., Bock, D. D. & Scott, K. *Drosophila* gustatory projections are segregated by taste modality and connectivity. Elife 11, e78110 (2022).

Jaeger, A. H. et al. A complex peripheral code for salt taste in *Drosophila*. Elife 7, e37167 (2018).

Macpherson, L. J. et al. Dynamic labelling of neural connections in multiple colours by trans-synaptic fluorescence complementation. Nat Commun 6, 10024 (2015).

Nässel, D. R. Leucokinin and Associated Neuropeptides Regulate Multiple Aspects of Physiology and Behavior in *Drosophila*. Int J Mol Sci 22, 1940 (2021).

Shiu, P. K., Sterne, G. R., Engert, S., Dickson, B. J. & Scott, K. Taste quality and hunger interactions in a feeding sensorimotor circuit. eLife 11, e79887 (2022).

Walker, S. R., Peña-Garcia, M. & Devineni, A. V. Connectomic analysis of taste circuits in *Drosophila*. Sci. Rep. 15, 5278 (2025).